# The short timescale variability of the oxygen inventory in the NE Black Sea slope water

Alexander G. Ostrovskii[1], Andrey G. Zatsepin[1], Vladimir A. Solovyev[1], Dmitry M. Soloviev[1,2]

[1]Shirshov Institute of Oceanology, Russian Academy of Sciences, 36, Nahimovskiy prospekt, Moscow, 117997, Russia
5  [2]Marine Hydrophysical Institute, Russian Academy of Sciences, 2, Kapitanskaya str., Sevastopol, 299011, Russia

*Correspondence to*: Alexander G. Ostrovskii (osasha@ocean.ru)

**Abstract.** Warm winters have recently become common over the Black Sea, leading to the risk of shoaling oxygen penetration. The insufficient supply of oxygen to the near-bottom layer may result in a decrease in faunal abundance. However, there is a lack of data on the temporal variations in oxygen throughout the water column over short timescales of 10  hours to weeks. In this paper, new observations over the upper part of the NE Black Sea continental slope are presented. Regular, frequent measurements were performed using a moored profiler from January to early March 2016. The profiling allowed for direct observations of the temperature in the cold intermediate layer (CIL), pycnocline structure, current velocity, and oxygen stratification and in particular, the depths of hypoxia onset. The average local oxygen inventory below a depth of 30 m was 24.9 mol m$^{-2}$. Relatively high/low oxygen inventory cases were related to the thin/thick main pycnocline, which 15  was associated with the onshore/offshore location of the Black Sea Rim Current. The pycnocline hindered the vertical transport of oxygenated water to the CIL. The vertical displacements of the hypoxia onset depth ranged from 97-170 m, while the shelf edge depth in this region usually ranged from 90-100 m. Intermittently, the hypoxia boundary depth fluctuated on two time scales: ~17 h due to inertial oscillations and 3-7 days due to current meanders and eddies.

## 1 Introduction

20     The Black Sea is a semienclosed basin with very limited water exchange with the Aegean Sea through the Turkish Straits (the Bosporus and Dardanelles). In the Black Sea, the permanent pycnocline separates the saline waters of Mediterranean origin from the upper layer diluted with fresh water due to precipitation and river discharge (for most recent research on Black Sea thermohaline properties, the long-term trends and variations, see Miladinova et al., 2017). The upper layer of the Black Sea is sensitive to climatic fluctuations and global warming (for example, refer to Stanev et al., 1995; 25  Belokopytov, 2011; Capet et al., 2012; Dorofeev et al., 2013; Dorofeev, Sukhikh, 2016).

The Black Sea is prone to natural hydrogen sulfide contamination, and oxygen-dependent life forms only exist in the top layer of the Black Sea, which has a thickness of less than 200 m, and this top layer is the most susceptible to atmospheric impacts (Oguz et al., 2003; Capet et al., 2004; Mikaelyan et al., 2013; Capet et al., 2014; Stanev et al., 2014). In the eastern Black Sea, the ecosystem stress is aggravated by the fact that the shelf is narrow (Fig. 1); thus, the benthic habitat is smaller

than the total area of the eastern Black Sea. In general, factors that provide Black Sea ecosystem resilience to anthropogenic and climatic effects are weaker than those in other marginal seas adjacent to the European continent (Kideys, 2002).

The sea is ventilated in the winter when convection and vertical turbulent mixing move the cold surface water deeper into the pycnocline (refer to Ivanov et al., 2001 and corresponding references). The vertical mixing enables the top layer to achieve the maximum thickness and minimum temperature (Stanev et al., 2003). After the winter, a seasonal thermocline is formed above the least heated layer termed the cold intermediate layer (CIL). The seasonal thermocline blocks vertical turbulent mixing and to a considerable extent, isolates the CIL from atmospheric forcings. For this reason, the thermohaline properties below the seasonal thermocline retain the "memory" of wintertime cooling. Originally, the CIL was defined by a temperature of less than 8°C (Ovchinnikov, Popov, 1987; Konovalov et al., 2005) and a temperature range of 8-8.6°C due to warming (Stanev et al., 2013). The CIL contains most of the thermocline over the Black Sea (Oguz et al., 1991). In the NE Black Sea, the CIL core defines the top of the strong halocline (Yakushev et al., 2005; Ostrovskii, Zatsepin, 2016). In this region, the upper part of the permanent pycnocline is partly supported by the vertical temperature gradient, while the lower part of the pycnoline is supported by only the halocline.

Cooling and convective mixing processes that occur in winter in the Black Sea supply oxygen to the top part of the permanent pycnocline, where turbulence rapidly decays (Gregg, Yakushev, 2005; Zatsepin et al., 2007). Gregg and Yakushev (2005) stressed that understanding the origin of the CIL has been hampered by the scarcity of winter observations. Detailed comprehensive studies, including the microstructure measurements and sampling in the center of the cyclonic gyre in anomalously cold winter conditions, were completed in March 2003 (Gregg, Yakushev, 2005). The freezing air outbreaks drove convection that cooled the surface mixed layer to 6.1°C and deepened the layer to 40 m, which directly ventilated the upper 80% of the CIL. The concentrations of dissolved oxygen were ~350 $\mu$mol L$^{-1}$ in the mixed layer and rapidly decreased to 70 $\mu$mol L$^{-1}$ at the base of the CIL, which is 9 m below the mixed layer. These observations are unique, and we assume that observations such as these have not been repeated.

During the winter in the NE Black Sea, the levels of dissolved oxygen, [O$_2$], are relatively high at the CIL upper boundary, e.g., Yakushev et al. (2005) presented profiles that indicated [O$_2$] ≈ 250 $\mu$mol L$^{-1}$ at a depth of 80 m where the temperature dropped below 8°C. The dissolved oxygen decreased with depth in the CIL. The oxycline extended from $\sigma_\Theta \approx 14.5$ kg m$^{-3}$ to $\sigma_\Theta \approx 15.75$ kg m$^{-3}$, where levels of [O$_2$] become undetectable (<5 $\mu$mol L$^{-1}$). This region was termed the suboxic layer (SOL) by Murray et al. (1989). For instrumental observations using dissolved oxygen sensors, a more appropriate definition of SOL would be the layer in which [O$_2$] < 10 $\mu$mol L$^{-1}$ (Stunzhas et al., 2013). According to Glazer et al. (2006), the SOL is 15-30 m thick in the eastern Black Sea. Hydrogen sulfide, [H$_2$S], remains undetectable (<1 $\mu$mol L$^{-1}$) below the SOL boundary unless the water density $\sigma_\Theta$ exceeds 16.02-16.06 kg m$^{-3}$ (Glazer et al., 2006).

Since 2010, data from the ARGO floats equipped with oxygen sensors became available for the Black Sea. The studies based on these data enabled quantitative assessments of the differences between the vertical structure in the central part of the Black Sea and the vertical structure in the periphery of the sea (Stanev et al., 2013). The oxygenated layer in the upper ocean in the central basin was approximately two times thinner than that in the coastal zone, and the temporal

variations in thickness were very small (approximately 20 m). Conversely, the depth of the dissolved oxygen isoline 5 μmol $L^{-1}$ underwent changes of more than 75 m within several weeks of the ARGO float travel time. Numerous strong, upward bursts of the low oxygen layer that reach 50 m in the deep basin and 75 m in the coastal zone were observed, most likely due to mesoscale processes (Stanev et al., 2013). The long-term trends assessed with the ARGO float data and the old archive data that pertained to marine and laboratory measurements demonstrated that the oxygen inventory had been decreasing in the Black Sea since the second half of the 20[th] century (Capet et al., 2016). These trends were apparently exacerbated by global warming, which can contribute to the elevation of the hydrogen sulfide boundary.

Capet et al. (2016) emphasized that the following objectives are important in the present stage of Black Sea research: determining the extent to which the shoaling of the oxygen penetration layer may entrain the shoaling of the hydrogen sulfide onset depth, establishing a system for the continuous monitoring of the oxygen inventory and CIL ventilation, clarifying and quantifying the diapycnic and isopycnic ventilation processes at the periphery of the basin, including mesoscale dynamic structures, and understanding how the intensity of mixing is related to the Rim Current dynamics.. To achieve these goals, Capet et al. (2016) proposed using an observation network of moored and drifting profilers equipped with oxygen and hydrogen sulfide sensors.

Unfortunately, ship-based monitoring alone cannot address low oxygen dynamics on short timescales or during episodic low oxygen events (Friedrich et al., 2014). Additionally, most of the ARGO floats profile the water column at five or ten-day intervals, and such data could not be used to evaluate the typical time scales of hours-to-days of oxygen inventory fluctuations. Even if the ARGO floats are programmed to perform profiling once per day (e.g., Stanev et al., 2017), this timing would not be sufficient for observing daily changes. Since the Black Sea ecosystem responses depend on the frequency, duration, and severity of hypoxia events, more frequent sampling of oxygen is required. For the onset of hypoxia at the Black Sea Crimean shelf, Jessen et al. (2017) used a threshold value [O2] = 63 μmol L−1 (Middelburg, Levin, 2009), which indicates the low oxygen content conditions that begin affecting the fauna, the structures of animal communities, and the functions of the marine ecosystem.

To study oxygen dynamics over short timescales, we deployed a mooring with a profiler Aqualog (Ostrovskii and Zatsepin, 2016) with a high-accuracy oxygen sensor at the NE Black Sea periphery from January to March 2016 (Fig. 1). The mooring site was located in the top part of the continental slope near Gelendzhik Bay. The NE Black Sea region near Gelendzhik is far from the influence of the Bosporus Strait's input and major river inflows. The vertical hydrological and hydrochemical structures in this region reflect "integrated" changes at timescales longer than the seasonal scale (Yakushev et al., 2005). Data from regular sampling expeditions into the NE Black Sea from 1989 to 2005 have been used to study the hydrochemical structure of the water column during winter and summer (Yakushev et al., 2005). However, the short timescale processes in the NE Black Sea slope water and in particular, the variability in the oxygen stratification at timescales of hours to weeks were poorly known because of the lack of observational data until the deployment of our profiler mooring.

The Aqualog profiler mooring site was located near the Rim Current, which encircles the entire Black Sea basin. In this region, the ventilation processes in the top layer are much more diversified than those that occur in the open sea. In particular, the meanders of the Rim Current and the mesoscale eddies cause a high amplitude (> 50 meters) and relatively rapid (on the scale of several days) vertical oscillations in the pycnocline and oxycline (Ostrovskii et al. 2010; Zatsepin et al. 2013). Near-bottom Ekman currents are also generated (Ostrovskii and Zatsepin, 2016; Elkin et al., 2017). Wind-induced upwellings (Zatsepin et al., 2016) sometimes initiate a local, short-term redistribution of the oxygen concentration over the shelf and continental slope.

The local processes in the NE Black Sea edge are of limited duration. The impact of local processes on oxygen dynamics as well as the temporal scales of the oxygen inventory fluctuations must be described more thoroughly, typical vertical distributions of oxygen must be identified, observed vertical distributions must be explained, and any gaps in our understanding of the effects of oxygen fluctuations on the sea ecosystem must be identified. These problems can be addressed by analyzing the time series of vertical high-resolution, multivariable datasets. The time resolution of the profiling data must be sufficiently high to capture any transient processes that may have a long-term effect on the ecosystem (Jessen et al., 2017).

Below, we consider these issues by analyzing a unique dataset that describes the vertical profiles of the oxygen content, temperature and salinity of water, and the speed and direction of currents, as well as the acoustic backscattering of suspended matter. These profiles were obtained during winter, when the maximum amount of the oxygen inventory in water is achieved by the frequent cycling of the Aqualog profiler between the near-surface sea layer and the near-bottom layer below the hydrogen sulfide boundary.

## 2 Methods

Local oxygen inventory measurements were obtained on the moored buoy station at 44°32.372′N 37°54.868′E near Gelendzhik Bay in the NE Black Sea (Fig. 1). The shelf is normally shallow, at less than 100 m deep, with a maximum width of 4 km in this region. The mooring was deployed at approximately 220 m and operated from January 1 to March 6, 2016.

The mooring was equipped with the Aqualog profiler (Ostrovskii, Zatsepin, 2016). The Aqualog was programmed to cycle up and down every 2 hours to obtain 12 depth profiles on ascending tracks and 12 depth profiles on descending tracks, for a total of 24 profiles of daily multivariable data. The parking depth of the profiler was 205 m, and the upper profiling depth was 25 m. Each profiling cycle began from the parking depth at an exact UTC hour. After ascending to the upper profiling depth in approximately 14 min, the profiler stopped for 20 min and then descended to the parking depth for approximately 13 min. Several profiler faults occurred at the beginning of the observations, which resulted in data loss for a total of four days. Beginning on January 9, the profiler worked smoothly until it was recovered. Sometimes when the mooring line was inclined due to a particularly strong ocean current, the mooring line was below 25 m while climbing up the upper stopper, and consequently, the profiler stopped in that location. Hence, for the oxygen inventory calculations below, the upper depth limit was chosen as 30 m.

The Aqualog sensor suite included the CTD 52-MP CTD temperature, conductivity and pressure probe with an SBE43F fast oxygen sensor and an acoustic Doppler current meter Nortek Aquadopp 2 MHz. The CTD and dissolved oxygen data were obtained every 2 s. Soon after recovery, the SBE 52-MP CTD probe with the SBE43F sensor was sent to the manufacturer for calibration. All sensors were in very good condition and demonstrated the smallest possible drift. The modified calibration coefficients were considered during the processing of the measurement data. The data employed in our studies of currents comprised the data obtained during the ascent of the profiler. The Aquadopp acoustic current meter was mounted in the top part of the apparatus, and the current measurements in the ascending sections were obtained in undisturbed waters. The current meter sampling frequency was 4 Hz. To suppress the measurement noise, the current meter data were smoothed by the running mean using a fixed window length of 51 samples, which corresponds to a vertical scale of approximately 2.5 m. After data recovery, we obtained 1490 sets of high-resolution multivariable profiles.

To assess the favorable conditions for developing turbulent mixing, Richardson number $Ri$ estimates are applied. The linear theory predicts the development of the shear instability of stratified fluid when the $Ri$ is less than the critical value, which is equal to $Ri = 0.25$. The value $Ri < 1$ indicates the possibility of flow instability (Orlanski and Bryan 1969).

The gradient Richardson number was computed as follows:

$$Ri = N^2 / |\Delta \boldsymbol{U} / \Delta z|^2 \tag{1}$$

where $N^2 = g/\rho_0 \, (\Delta\sigma_\theta / \Delta z)$ is the Brunt–Väisälä frequency, $\Delta\sigma_\theta$ and $\Delta\boldsymbol{U}$ denote the difference in the water density and the difference in the horizontal current velocity, respectively, between adjacent depth bins with the thickness $\Delta z = 4$ m.

The turbulence rapidly decays in the pycnocline (Gregg, Yakushev, 2005, Zatsepin et al., 2007) so that the vertical mixing usually cannot bring a substantial amount of the oxygenated water deeper into the pycnocline. Hence, for average vertical distribution computations of $Ri$ as well as dissolved oxygen, the pycnocline core depth is worth using as a reference depth. In such a case, the modified depth is defined as follows:

$$\zeta = z - D^j_{\nabla_z\sigma_\theta} + < D_{\nabla_z\sigma_\theta} >, \tag{2}$$

where $z$ is the original depth, $D^j_{\nabla_z\sigma_\theta}$ is the depth of the maximum density gradient $\nabla_z \sigma_\theta = \Delta\sigma_\theta / \Delta z$ in the pycnocline for profile number $j$, and the angle brackets denote the time average. Notably, the time average $< D_{\nabla_z\sigma_\theta} >$ is added in (2) simply to ensure that the modified depth $\zeta$ is positive.

To justify the vertical displacement time scales of the oxygen concentration isolines, the power spectrum estimation was employed by using the multitaper method (MTM). According to the MTM approach, the statistically independent spectral estimates of a signal are obtained using the set of orthogonal weight sequences. The MTM method advantages were discussed in detail by Thomson (1982) and Wunsch (1999).

**3 Results**

**3.1 Background observations**

During the survey, the flow variability could be described as that due to a meandering near-coastal jet and propagating mesoscale and submesoscale eddies. Notably, the profiler mooring was located in the Rim Current zone on the periphery of the basin-wide cyclonic gyre, where the NW direction flow prevailed. The filaments and patches of suspended particles along the Caucasus coast of the Black Sea were observed using data from the Suomi National Polar-orbiting Partnership satellite visible infrared imaging radiometer (Fig. 2). The filaments traced the current meanders and mesoscale eddies. In January-February 2016, the anticyclonic eddies were 30-40 km in diameter, i.e., approximately 2 times the local Rossby radius of baroclinic deformation (Zatsepin et al., 2011). These eddies were long lived (approximately 1 month or more), and therefore, the eddies were quasi geostrophic in nature. The eddies moved northwestward, producing a water exchange between the shelf and deep sea. The narrow continental slope and shelf region was a dissipation zone for the dynamic structures and flows. Between the anticyclones and coast, there were submesoscale cyclonic eddies of smaller space-time scales up to several kilometers and several days. According to the Aqualog current meter data, the strengthening events of the NW transport (shown by the red color in upper Fig. 3) persisted for up to 7 days with the current speed occasionally reaching 45 cm/s in the upper sea layer.

The cross-shelf component of the flow velocity was much weaker than that directed along the shelf (lower Fig. 3). The flows toward the deep basin were intermittent and weak in the upper sea layer. During the survey, there were no upwelling events that would lead to a significant increase in the water exchange between the sea shelf zone and deep sea.

Occasionally, over the mooring site in the upper sea layer, the SE flow appeared for 1-2 days, but the speed of the flow did not exceed 20 cm/s. In such cases, the station was on the eastern periphery of the anticyclonic vortices, where the orbital velocity was southeastward (shown by the blue color in upper Fig. 3). During the second half of the survey, southeastward transport was observed more often. The SE current was noticeable in the pycnocline between isopycnals $\sigma_\Theta$ from 14.5 to 16 kg m$^{-3}$.

On a shorter timescale, near-inertial motions emerged. The inertial period at the latitude of the Aqualog mooring site is approximately 17.06 hours. The inertial signal was strong from January 16 to 25 and from February 1 to 10. The inertial oscillations could be induced by a local forcing or could arrive as internal-gravity waves to the sea edge from the deep sea. The nature of the inertial oscillations observed by using the Aqualog profiler mooring is beyond the scope of this study.

Finally, as background observations, the weather conditions should be briefly mentioned here. According to the Gelendzhik weather station of the Russian Hydromet Service, the weather was warm in January-March 2016 (Melnikov et al., 2018). Several cold days occurred from January 1-4 and January 23-26 when the air temperature was freezing. The monthly average air temperature was 4.08°C in January, 8.27°C in February, and 9.19°C in March.

**3.2 Oxygen dynamics**

Direct measurements showed that the vertical distribution of oxygen underwent significant fluctuations from January-March 2016 (Fig. 4). Noticeably, the temporal variations of the dissolved oxygen were usually locked to the vertical

displacements of the isopycnal surfaces. The upper sea layer above the isopycnal $\sigma_\Theta = 14$ kg m$^{-3}$ was usually well-mixed. The dissolved oxygen in the uppermost part of the water column was from 280 µmol L$^{-1}$ in January to 310 µmol L$^{-1}$ at the beginning of March (Fig. 4).

From January to March 2016, the average oxygen inventory $I$ in the Aqualog station region below 30 m in the sea was $\bar{I}_{30} = 24.9$ mol m$^{-2}$. The value of $I$ fluctuated highly on two scales: the near-inertial scale and over three to seven days (refer to Fig. 5). The standard deviation of the $I_{30}$ time series was 3.5 mol m$^{-2}$.

To evaluate the total average oxygen inventory estimate, the average oxygen content in the 30-m-thick near-surface sea layer is assumed to be equal to the value $[O_2] = 270$-$300$ µmol L$^{-1}$ observed at a depth of 30 m, where the Aqualog profiler instrumental measurements were obtained. Then, the total average oxygen inventory for the entire water column would be the value $\bar{I}_{30}$ increased by approximately 8-9 mol m$^{-2}$, where $\bar{I}_{total} = 32.9$-$33.9$ mol m$^{-2}$.

The oxygen inventory distribution with depth varied substantially with time. In particular, the distribution was extremely small ($I_{\sigma 14} < 3.5$ mol m$^{-2}$) for approximately 11% of all profiles in the layer below the $\sigma_\Theta = 14$ kg m$^{-3}$ isopycnal (Fig. 5). In these cases, the oxygen-rich and deep upper layer was separated by a narrow oxycline from the underlying waters, which were subjected to the lower oxygen content conditions (Fig. 6). The thermohaline stratification was characterized by a very sharp pycnocline (Fig. 7), $< D_{\nabla_z \sigma_\theta} > = 117.2$ m. The sample-averaged Brunt-Väisälä frequency in the region of the maximum density gradient amounted to $<N> = 0.041$ s$^{-1}$ with the median value med($N$) equal to 0.037 s$^{-1}$. The average vertical distribution of the oxygen inventory for $I_{\sigma 14} < 3.5$ mol m$^{-2}$ had a two-layered structure and a single-step profile (Fig. 8); the oxygen content decreased from approximately 280 to 10 µmol L$^{-1}$ across the oxycline with a maximum thickness of 40 m. Note that in mid-January 2016, the distance between the oxygen-rich zone and upper boundary of the suboxic zone, where $[O_2] < 10$ µmol L$^{-1}$, was occasionally less than 20 m (Fig. 4).

The presence of a well-mixed, deep top layer is confirmed by the acoustic backscatter distribution data, which was measured by the acoustic Doppler current meter Nortek. According to the estimated cross-correlation amplitudes of acoustic backscatter (not shown here), the suspended matter coherently fluctuated throughout the entire top layer. Under the pycnocline, the suspended matter slowly subsided at an average rate of 3.5 mm/s.

The profiles where $I_{\sigma 14} \geq 3.5$ mol m$^{-2}$ comprised approximately 89% of all profiles ($< D_{\nabla_z \sigma_\theta} > = 118.9$ m). In such cases, the oxygen slowly decreased with depth in the oxycline to a typical thickness of ~100 m (Fig. 8). In the second half of February, the thickness of the oxycline began to increase, and the oxygen inventory below 30 m in the sea decreased from the peak values of $I_{30} = 30$-$35$ mol m$^{-2}$ in late January or early February to $I_{30} = 21$-$25$ mol m$^{-2}$ in early March (Fig. 5).

As the changes evolved, the depths of the hypoxia onset, SOL, CIL and pycnocline fluctuated with large amplitudes with a period of near-inertial oscillation (Fig. 9). The 3-7 day fluctuations were mainly observed in the layer deeper than the $\sigma_\Theta = 14$ kg m$^{-3}$ isopycnal. In the second half of January and the first half of February, these fluctuations were slightly modulated by near-inertial oscillations with a period of approximately 17 hours. Intermittently, at timescales of 3-7 days, the depth of hypoxia onset fluctuated with an amplitude of ~50 m. The periods of such fluctuations were interleaved by several

days of quiet. The fluctuations were observed in the 100-150 m layer, i.e., too deep to be directly associated with storms on the sea surface.

The inertial oscillations of the hypoxia upper boundary vertical displacements were verified by the power spectrum estimates. The spectral density was calculated for the hypoxia onset depth time series of January 4-27 when there were no missing data (Fig. 10). The obtained spectrum has a large peak in the inertial frequency band, which is significant with a 95% probability. Most of the energy belongs to the lower-frequency variability; the spectrum rapidly grows and then flattens at timescales longer than 3.5 days. Admittedly, the time series with no missing data is slightly short for spectral density estimations on several day timescales.

According to our data, the hypoxia onset often appeared in the region of maximum density gradients in the pycnocline and occurred above the minimum temperature depth in the CIL until March 3. The CIL gradually eroded while remaining unventilated all winter in this region. The temperature increased from 8.45°C in early January to 8.6°C between late February and the beginning of March, when a new CIL emerged as a result of horizontal advection in the 40 m layer above the old CIL (Fig. 11). In this new CIL, a temperature of 8.35°C was recorded, which was the absolute minimum temperature for the entire duration of our observations. The new CIL was located above the hypoxia zone.

## 4 Discussion

Few studies of the oxygen distribution in the Black Sea periphery have mentioned short-term oxygen content fluctuations in the upper sea layer that extend down to the main pycnocline. The observations using drifting ARGO floats provided insight into the important role of mesoscale processes in both the central part of the basin and the Rim Current zone (Stanev et al., 2013). However, quantitative data on the short-term variability of the hypoxia layer depth and the suboxic zone depth are lacking. Mesoscale and submesoscale eddies, subsurface waves and upwelling events were known to interact in the shelf and slope zone at time scales of less than 10 days (Zatsepin et al., 2013).

The results obtained in this study highlight the basic time scales of the oxygen inventory changes in the NE Black Sea: the inertial period and the scale of 3-7 days. The short-term variations are superimposed on the intraseasonal change. The maximum total oxygen inventory can be as large as 42-43 mol m$^{-2}$ in February 2016, with a winter average of 33-34 mol m$^{-2}$. The latter value agrees with the annual average for this region (refer to Fig. 4 in (Capet et al., 2016)).

What were the hydrophysical conditions when both the pycnocline and oxycline became so thin during wintertime? Based on our data, these conditions can be described by strong NW currents (Fig. 12) in the top layer above the pycnocline. The maximum NW current speed sometimes exceeded 0.4 m/s. The current speed boost may occur when the Rim Current axis shifts toward the shelf in the region. When the Rim Current is pressed against the continental slope, the isopycnals are usually lowered by 40-60 m (Zatsepin et al., 2011).

A countercurrent in the SE direction was observed in the narrow pycnocline. The maximum speed (sometimes as high as 0.3 m/s) of the countercurrent was in the layer where the vertical density gradient attained the maximum values. This

countercurrent was observed in 2007 during the first experiment with a moored profiler in the Black Sea (Ostrovskii et al., 2010).

While the oxygen inventory was limited in the layer of $I_{\sigma 14} < 3.5$ mol m$^{-2}$ in the mooring station region, the vertical oxygen distribution in the top layer was uniform, and the stratification was nearly neutral. The median of the Richardson numbers med($Ri$) indicated that the condition $Ri < 0.25$, which was required for vertical turbulent mixing, may often develop in the top 100 m (Fig. 13). In the pycnocline, the median values of $Ri$ abruptly increased to 2-6. Under the pycnocline (down to 200 m depth), conditions for turbulent mixing at the 4 m vertical scale usually do not occur considering the estimated $Ri$ values. However, we cannot exclude that these conditions occur at smaller vertical scales.

In the deep basin, the short-term fluctuations of the hypoxia onset do not seriously endanger the ecosystem because the plankton floating at the isopycnic surfaces coherently fluctuate with the depths, and the oxygen isolines almost always move in accordance with the isopycnals. Along the NE Black Sea edge, the vertical displacements of the hypoxia zone pose a risk for the ecosystem over the continental slope. The pycnocline shrinkage to a thickness of 15-20 m near the shelf edge for several hours may create favorable conditions for entraining biota into the suboxic zone when an internal wave breaks near the sea bottom. This wave breaking is common when the vertical current shear considerably increases after the internal wave runs onto the continental slope or shelf. In the NE Black Sea, the maximum amplitude of short-term (7-8 minutes) subsurface waves on the shelf may be as high as 16 m (Bondur et al., 2018 (to be published)). In the NW Black Sea, intensive subsurface waves in the lower part of the pycnocline in the halocline were primarily observed in the continental slope area in autumn, winter and summer (Morozov et al., 2017). During these measurements in the main pycnocline in December 2012, vertical current profiles of the large internal waves were observed with vertical lengths of 50-60 m.

An abruptly fluctuating hypoxia onset in the shelf and slope sea zone may considerably affect the benthos. Note that the shelf edge in the NE Black Sea predominantly occurs in the depth range of 90-100 m; thus, the extensive shoaling at the hypoxia onset depth should enable penetration of low oxygen water into the outer shelf. The hypoxia events near the shelf edge can be observed based on our data for January 1-3, 2016. Ground species living at the top part of the continental slope adapt to oxygen fluctuations because these species should experience hypoxic conditions at regular intervals of ~17 hours.

## 5   Conclusion

A higher total content of oxygen that exceeds 34 mol m$^{-2}$ and sometimes exceeds 40 mol m$^{-2}$ in the NE Black Sea slope water column was observed in approximately 11% of the profiles collected during a warm winter between January to early March 2016. An oxygen inventory increases when the Rim Current axis is shifted closer to the shelf edge. Under these conditions, the top mixed layer is deep (110-130 m) and oxygen-rich (270-300 µmol L$^{-1}$ of oxygen on average), whereas both the thickness of the pycnocline and oxycline decrease to less than approximately 40 m.

Weakening of the Rim Current jet or the shift toward the open sea causes an expansion of the pycnocline, and consequently, a decrease in the local oxygen inventory (less than 33 mol m$^{-2}$). In the latter case, stable stratification in the

thick pycnocline tends to suppress vertical turbulent mixing from a depth of 30-40 m. The Rim Current that flows in the NW direction occupies the upper sea layer to the top of the pycnocline. The current speed vertical shear in the lower part of the current is too small to overcome stable stratification in the pycnocline, and therefore, turbulent mixing cannot reach the CIL. During a warm winter, the maximum gradient in the pycnocline is located above the CIL core; thus, the CIL cannot be more or less intensively ventilated. By the end of the warm winter, a new CIL may be formed due to the horizontal advection of the water above the old CIL. Water in this new CIL is more oxygen-rich at [O2] = 240-270 µmol $L^{-1}$ and colder (8.3–8.4°C) than in the old CIL. Considering that the new CIL is located above the hypoxia zone, essential changes in plankton migrations may occur. Some preliminary results on how the physical-chemical and behavioral properties drive the vertical structure of mesozooplankton distribution in the NE Black Sea were presented by Arashkevich et al. (2013). A more detailed study is currently underway.

In the wintertime, when the top layer of the Black Sea is intensively ventilated, the hypoxia onset depth may be as shallow as 97 m near the boundary of the sea shelf and continental slope, i.e., nearly above the shelf edge in the study area. Most likely, the life cycles of the near-bottom-layer species that live in the outer shelf deeper than ~95 m had to adapt to significant oxygen fluctuations in the water at two time scales of approximately 17 hours and 3-7 days. Studies of the biota in the boundary area of the Black Sea ecosystem are currently underway (Jessen et al., 2017).

A key question is how the oxygen stratification will change under conditions of continued global warming. Will climate change cause the 63 µmol $L^{-1}$ hypoxia zone to persist above the sea shelf edge and cause the benthos at the NE Black Sea shelf to enter hibernation in association with the low oxygen content of the near-bottom layer? In the worst case scenario, the broad shelf region south of the Kerch Strait with fishing grounds only 100 km to the northwest of our observational site would become the next environmental risk area.

*Acknowledgements*. Authors are thankful to P.A. Stunzhas for discussions about measurements of the dissolved oxygen in the Black Sea. This research was performed in the framework of Program of the Presidium of Russian Academy of Sciences No. 1-2-50 (0149-2018-0022) and supported in part by Russian Fund for Basic Research (project No. 17-05-00381).

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

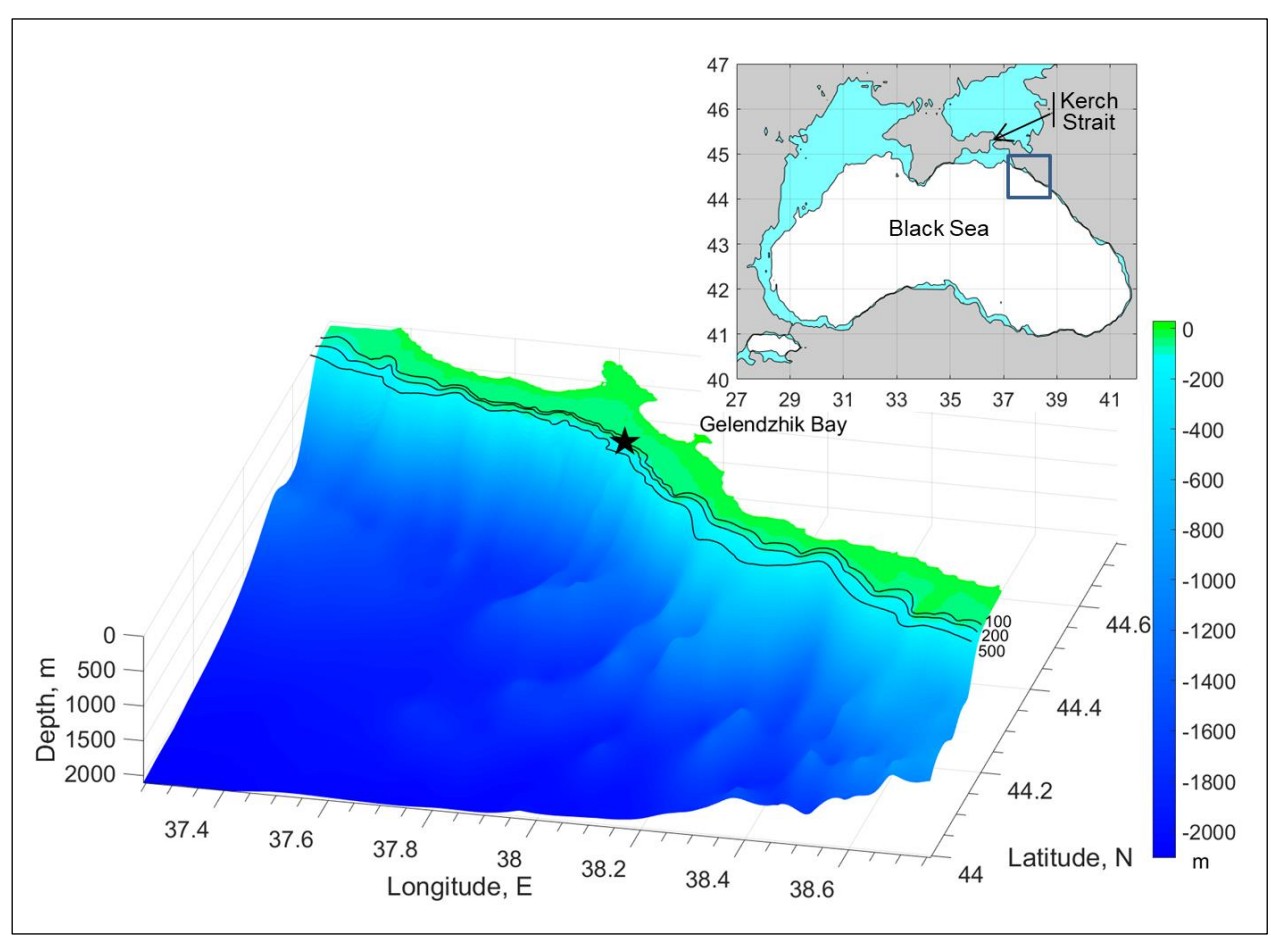

**Figure 1: Bottom topography of the NE Black Sea (based on European Marine Observation and Data Network bathymetry; http://www.emodnet-hydrography.eu/). The black star indicates the location of the Aqualog profiler mooring. Notice the narrow (width of only a few kilometers) and shallow (depth less than 100 m) sea shelf.**

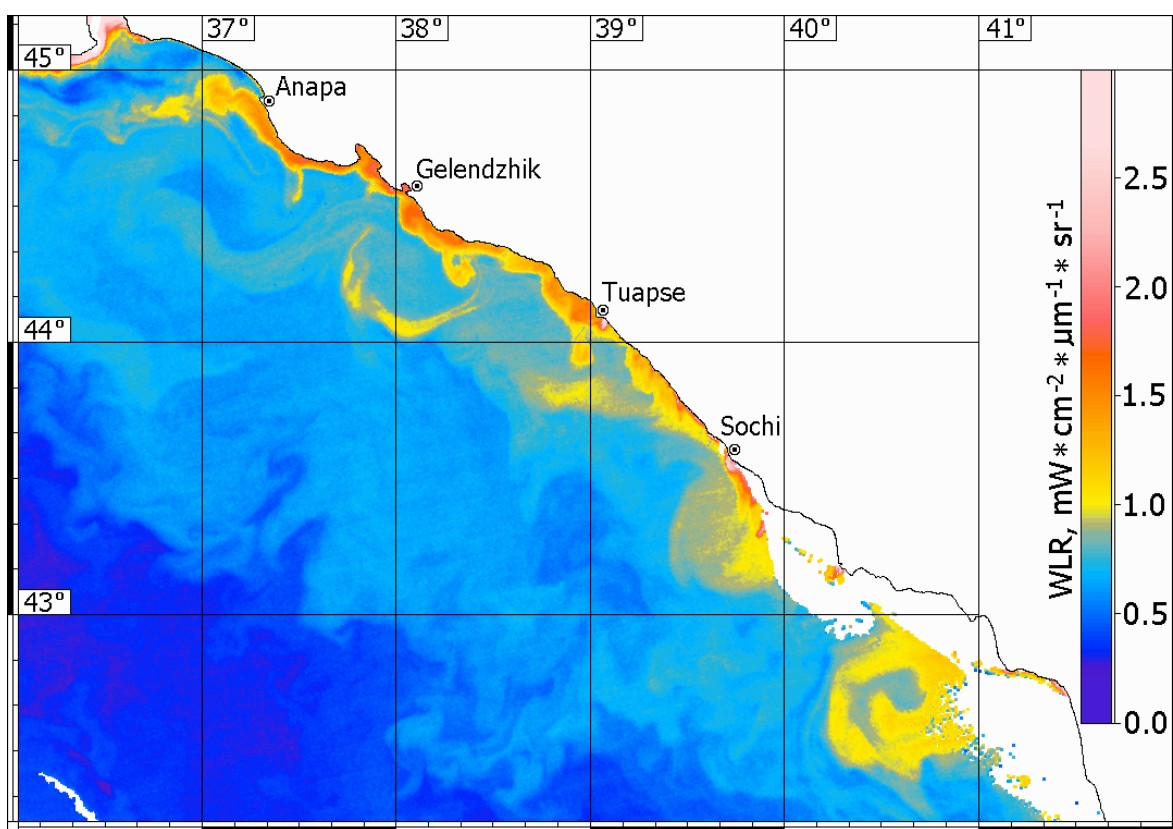

**Figure 2: The Suomi NPP Visible Infrared Imaging Radiometer Suite image of the water leaving radiance (WLR) at a wavelength of 0.551 mm over the NE Black Sea at 10:18 GMT on February 18, 2016 (data are courtesy of NASA at https://oceancolor.gsfc.nasa.gov/).**

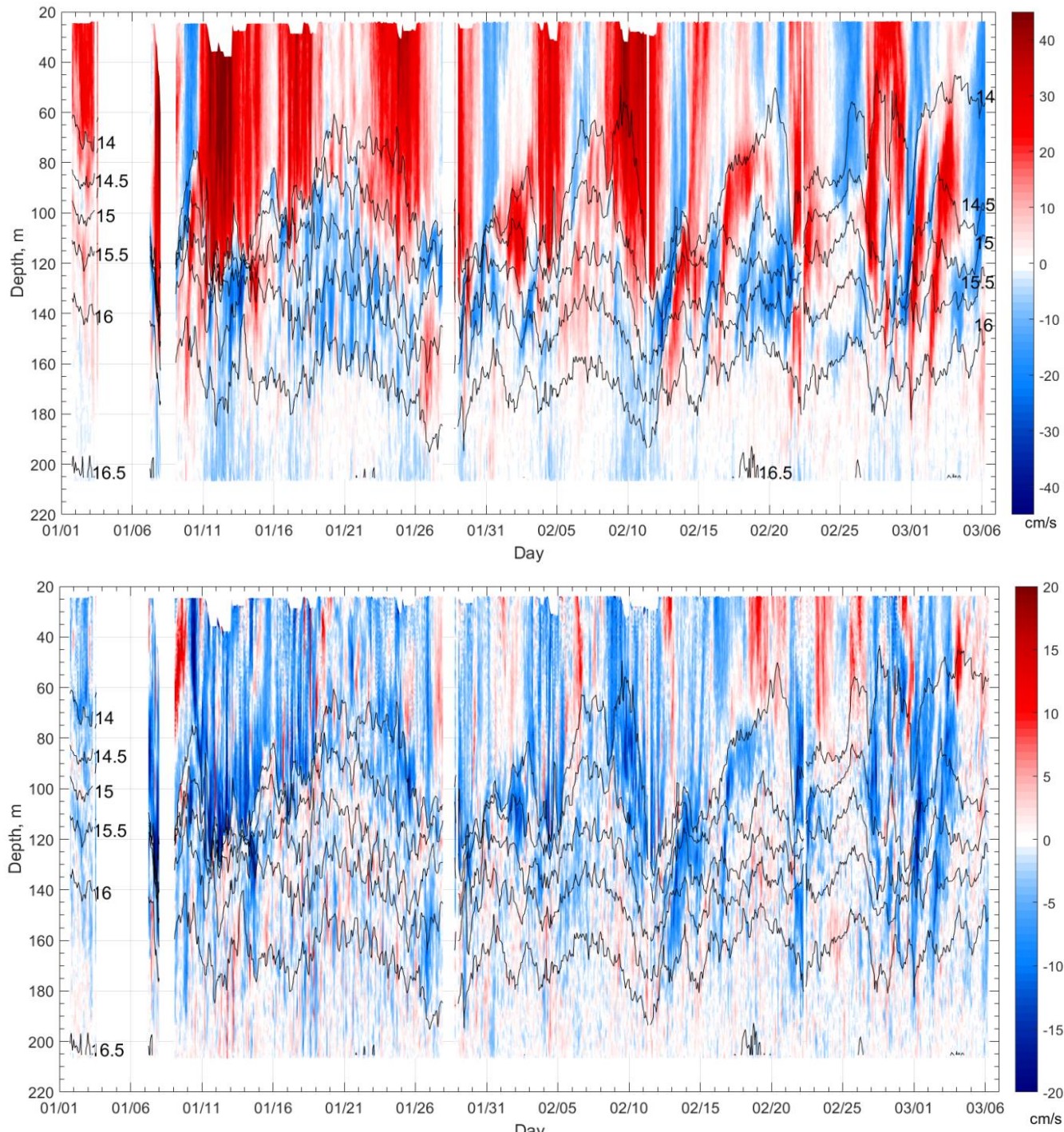

**Figure 3: Profiles of along-shore (upper panel) and cross-shore (lower panel) components of the current velocity, depicted as colored contours. The isopycnals $\sigma_\Theta$ are superimposed (black lines). Note that the positive direction of the along-shore axis is 50°, and the positive direction of the cross-shore axis is 320°.**

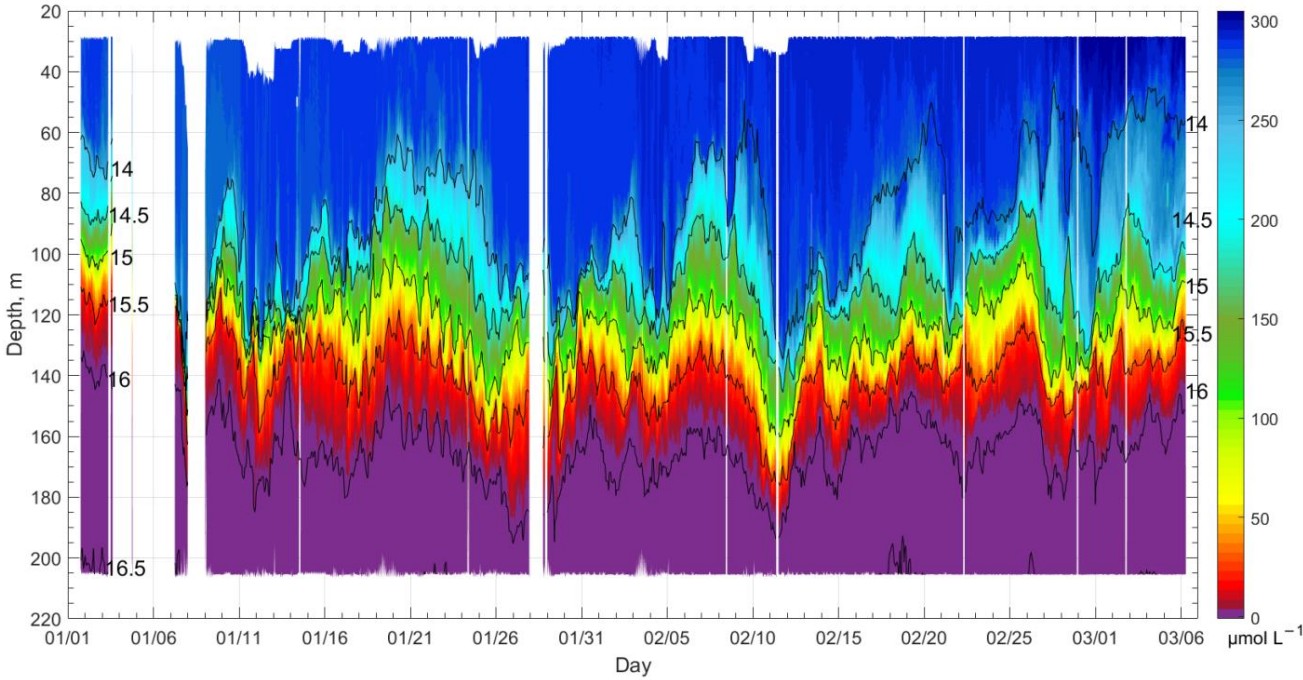

**Figure 4: Temporal evolution of the dissolved oxygen vertical distribution. The measurements were obtained using an SBE 43F sensor housed in an Aqualog moored profiler in the NE Black Sea (see Figure 1 for location). The black lines indicate isopycnals** $\sigma_\theta$.

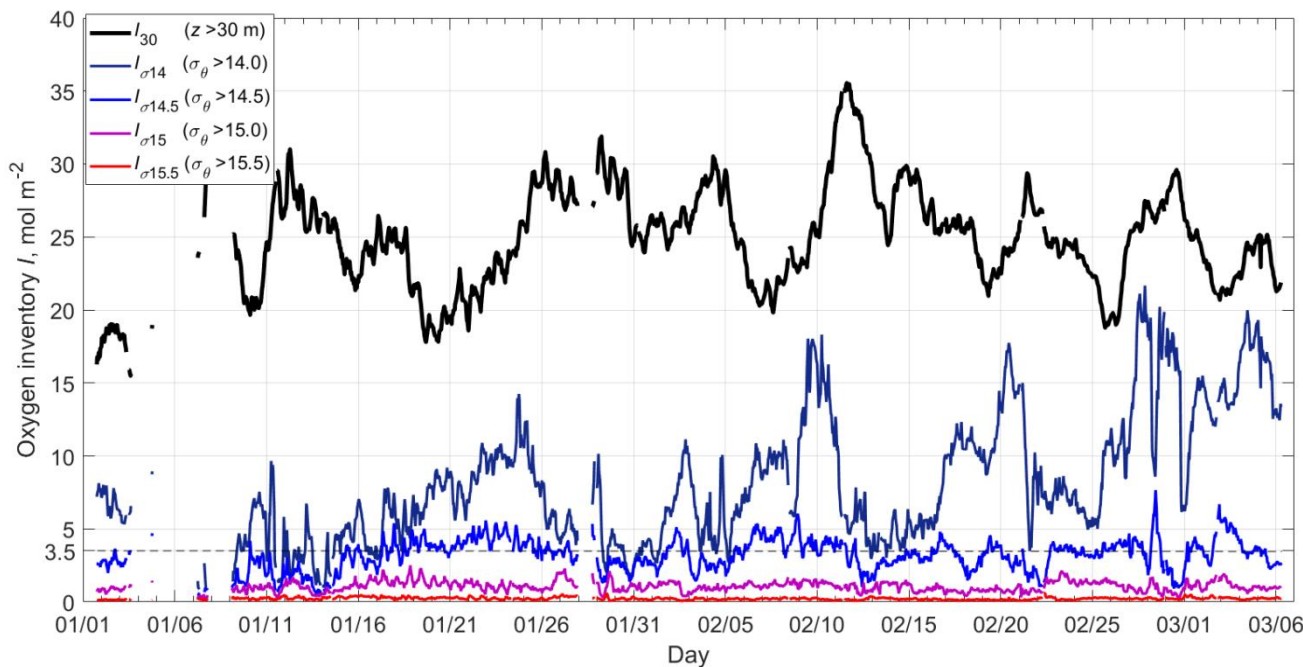

**Figure 5: The oxygen inventory *I* per the Aqualog profiler data acquired from January to March 2016. The green line is the oxygen inventory $I_{30}$ below 30 m in the sea. Other colored lines denote the oxygen inventories $I_{\sigma14, ..., \sigma15.5}$ in the layers below the respective isopycnal depths, as specified in the top left corner of the figure.**

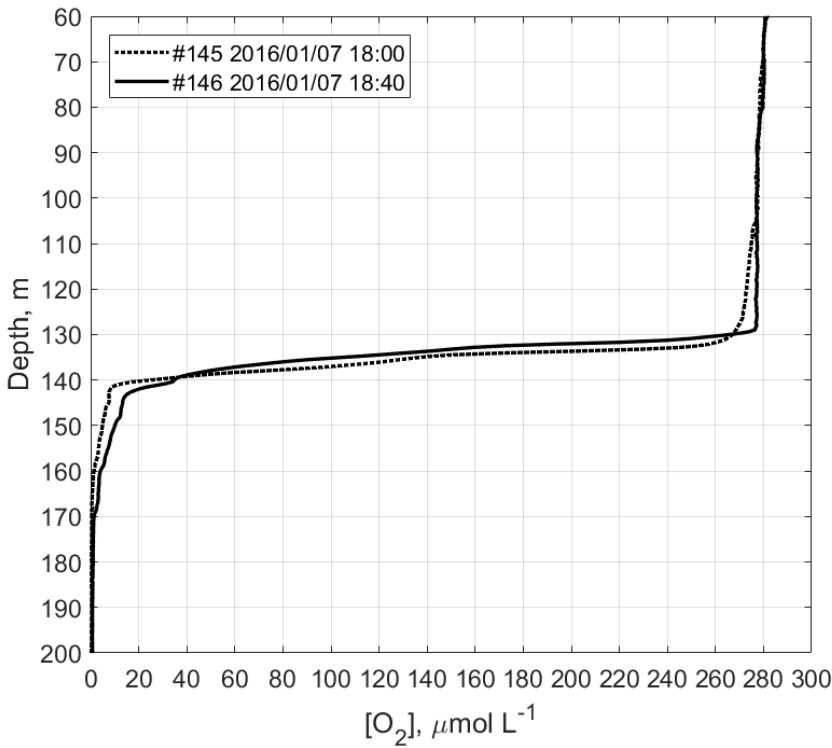

**Figure 6: Example series of five profiles $[O_2]^j(z)$, where $z$ is the depth and $j$ is the profile number, $j$ = 145 and 146, when the oxygen inventory below $\sigma_\Theta$ = 14 kg m$^{-3}$ isopycnal was extremely low $I_{\sigma14} < 3.5$ mol m$^{-2}$. The profile numbers, dates and times are shown in the top left corner. The odd-numbered profiles were taken when the profiling probe was moving up, and the even-numbered profiles were taken when the profiling probe was moving down. Due to the time-lag of the oxygen sensor, a profile obtained during the descent has slightly different curvatures compared with that obtained during the ascent, particularly at the upper and lower boundaries of the oxycline.**

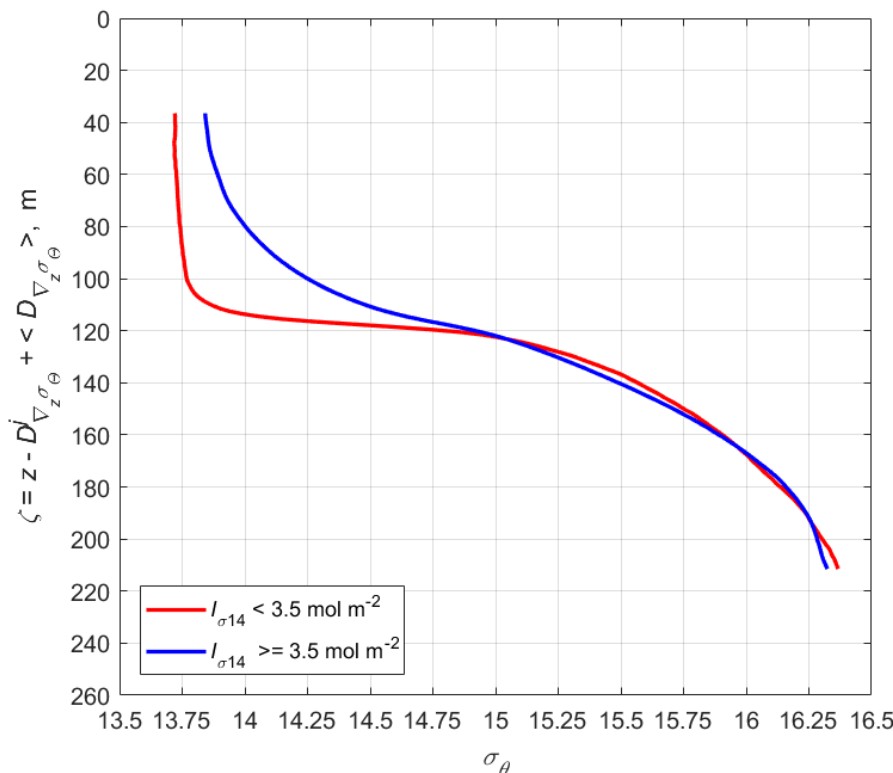

**Figure 7: Mean vertical profiles of potential density *versus* distance from the depth of the maximum density gradient in the pycnocline. The mean values were calculated for an ensemble that consists of $M$ individual profiles $\sigma_\theta^j(z)$, where $z$ is the depth and $j = 1, \ldots, M$; each are vertically displaced from the depth of the maximum density gradient in the pycnocline $D_{\nabla_z\sigma_\theta}^j$: $\zeta = z - D_{\nabla_z\sigma_\theta}^j + \langle D_{\nabla_z\sigma_\theta} \rangle$, where $\nabla_z\sigma_\theta = \Delta\sigma_\theta/\Delta z$ and the angle brackets denote the time average. The red curve represents the mean profile for the cases where $I_{\sigma14} < 3.5$ mol m$^{-2}$. The blue curve is the mean profile for the cases where $I_{\sigma14} \geq 3.5$ mol m$^{-2}$,**

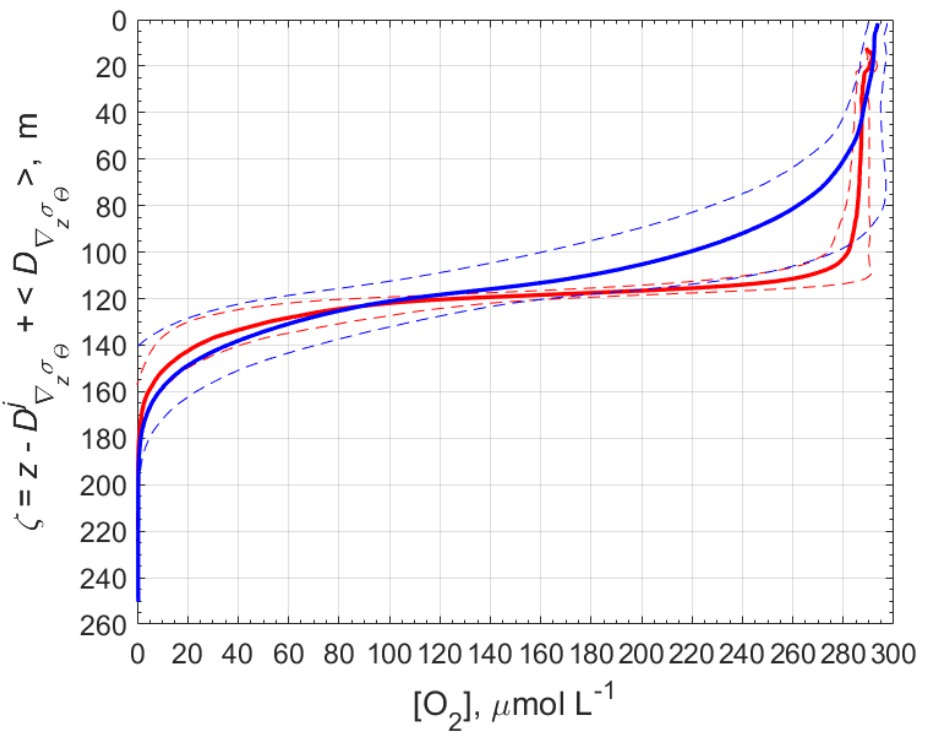

**Figure 8: Mean profiles of dissolved oxygen content for $I_{\sigma 14} < 3.5$ mol m$^{-2}$ (red curve) and $I_{\sigma 14} \geq 3.5$ mol m$^{-2}$ (blue curve).**

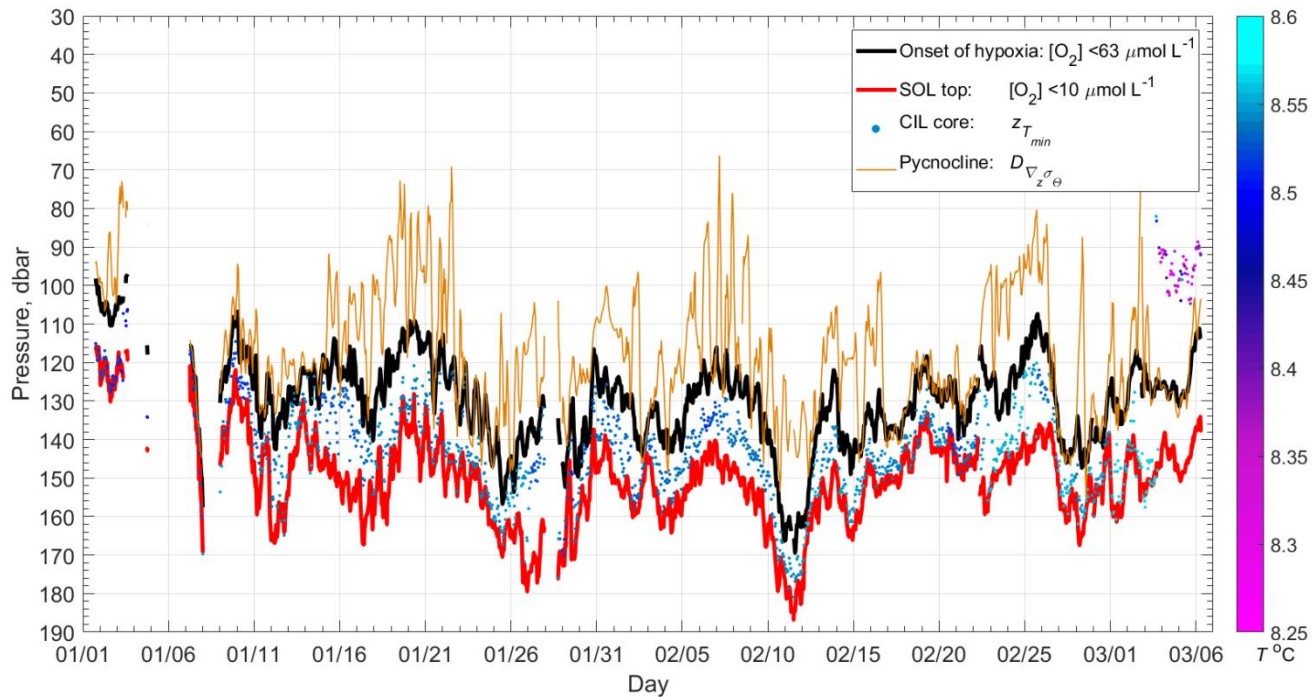

**Figure 9: Depths of the hypoxia onset, suboxic layer, and maximum vertical gradient of density and the minimum water temperature (regarding the colored spots, refer to the color scale to the right of the figure).**

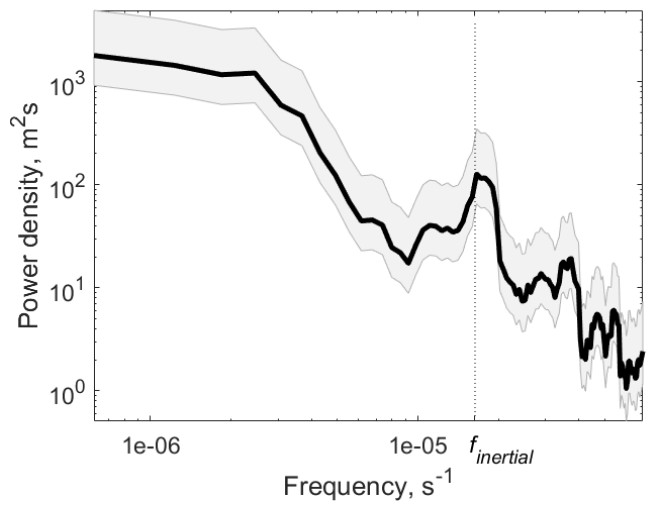

**Figure 10: Spectral density of the hypoxia onset depth for January 4-27, 2016. The 95% confidence interval is shown in gray.**

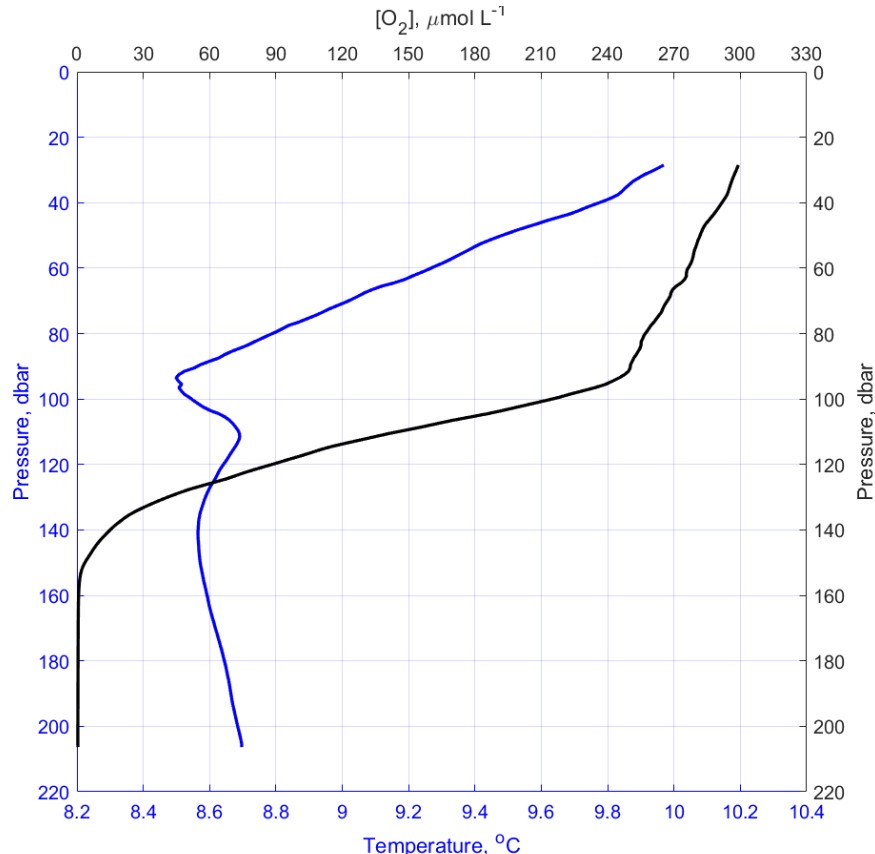

**Figure 11: Mean profiles of water temperature and oxygen concentration during the period from 16:00 UTC 3.3.2016 until 07:00 UTC 6.3.2016. Note the minimum temperature in the new CIL at the depth range of 70-110 m.**

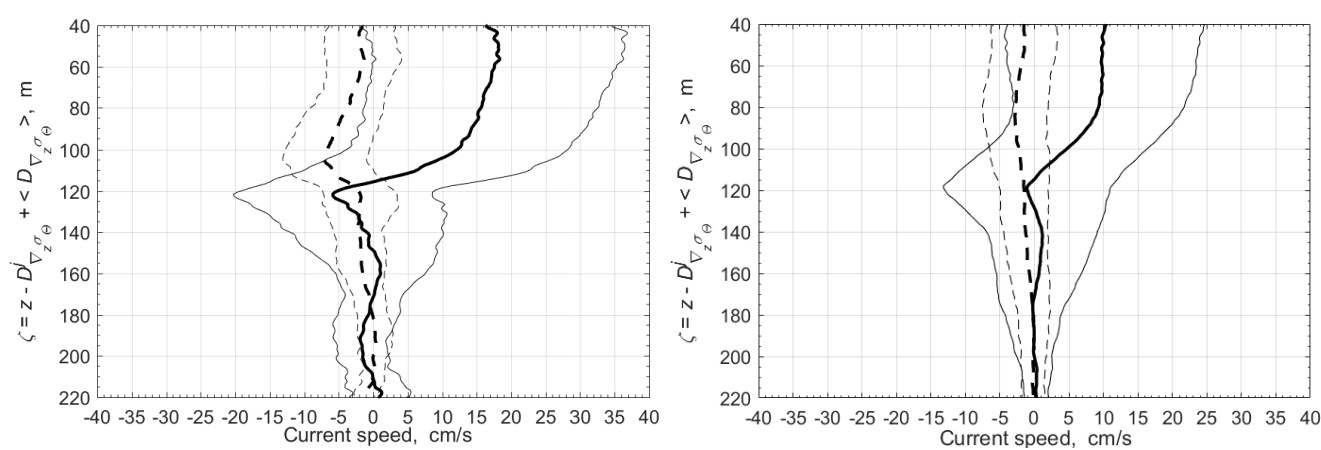

**Figure 12: Depth profiles of along-shelf (NW direction is positive; solid curve) and cross-shelf (NE direction is positive; dashed curve) current speed components, which are derived from the ensemble of profiles when $I_{\sigma 14} < 3.5$**

mol m$^{-2}$ (left) and when $I_{\sigma14} \geq 3.5$ mol m$^{-2}$ (right) from January-March 2016. The thick curves show the mean profiles, while the thin curves indicate standard deviations.

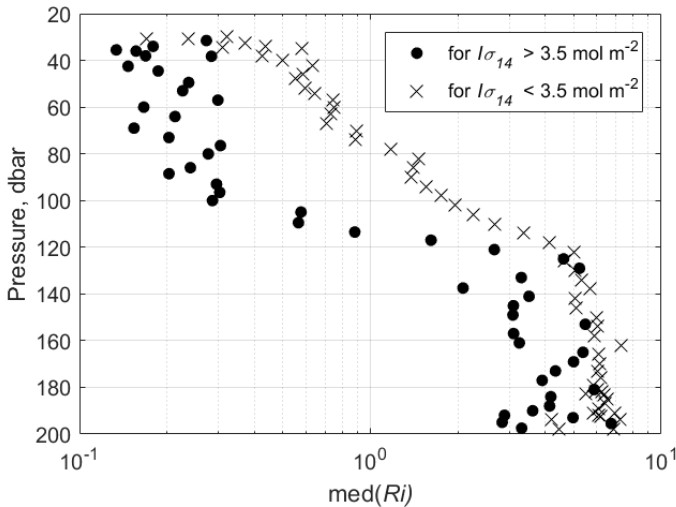

**Figure 13: Vertical profiles of the median values of the Richardson numbers, which are estimated based on two data ensembles: for $I_{\sigma14} < 3.5$ mol m$^{-2}$ and $I_{\sigma14} \geq 3.5$ mol m$^{-2}$.**