# Peer review of "The short timescale variability of the oxygen inventory in the NE Black Sea slope water"

_Ocean Science, 2018_

## Referee Comment (RC1) · S. Konovalov (Referee) · 17 Sep 2018

**S. Konovalov (Referee)**

sergey\_konovalov@yahoo.com

Received and published: 17 September 2018

**General comments**

The authors account for an extremely interesting issue of short-time variations in the distribution of oxygen in the Black Sea. They present and analyze the oxygen dynamics on the timescale of hours-to-days. This has never been done for the absence of observational data until now. Both the applied technique and acquired data are unique making the work exceptionally interesting and important. The authors have identified oscillations in the distribution of oxygen and other oxygen-dependent features, for example the onset of hypoxic conditions, on the timescale of  $\sim$ 17 hours and about 5 days. They have concluded that this might be vitally important for benthic communities at the

shelf edge experiencing periodic abrupt loss of oxygen.

The authors have also traced variations in the stock of oxygen in the water column for about 2 months of their experiment. They have revealed the influence of lateral advection on inventory of oxygen in the cold intermediate layer due to late-winter ventilation.

As far as I see, all observational results and achieved conclusions are new and scientifically important. This makes me sure that the manuscript is worth been published, but after extended improvement. There are, unfortunately, two major disadvantages in this manuscript: its structure and English.

I will make almost no specific comments on English, as this is not my first language, but I definitely see that English must be improved. The current version will not only irritate readers, but also often mislead them. I see in some places that the text is written in Russian by English words making hardly possible to understand the very brilliant results and discoveries.

Where it comes to the structure, the manuscript needs it badly. The major issues are not listed at the end of introduction. Instead, it says that "this paper considers these issues by analyzing a unique data set that describes the vertical profiles of oxygen content, temperature and salinity of water, and speed and direction of currents, as well as acoustic backscattering on suspended 5 matter." And "these issues" mean anything and everything the authors do with their observational data in section 4 "Discussion". Only concluding remarks suggest a better-structured text with observations and discoveries on the oxygen dynamics connected to water dynamics and hydrologic processes. Still, periodic oscillations in the distribution of oxygen remain identified but explained. And the final paragraph suggest some conclusions on the low oxygen content in the near-bottom layer that come from nowhere but need to be discussed in the previous section.

Thus, my final general comment that the suggested manuscript is unique in data and discoveries, but it needs to be improved in its structure and English.

OSD
Specific comments.

Consider to change the title for "The oxygen dynamics at the shelf edge of the Black Sea on the timescale of hours-to-days".

Consider to delete the last sentence in the abstract because it is hardly the major and final result of this work.

Page 1, line 15: Unless I have missed it, any explanation or justification of the chosen depth of 30 meters has not been suggested in the text.

Page 2, lines 1-2: This sentence comes from nowhere. Suggest a reference or some justification.

Page 2, lines 5-6: CIL and other specific for the Black Sea features need an explanation or references.

Page 2, line 11: The referenced publication by Oguz et al. is good but it is about modeling. Look for the recent publication by Kubryakov and Stanichniy or recently published data by Yunev et al., Mikaelyan et al. based on observations.

Page 3, line 5: I do not see any reason for discussions of global warming in this manuscript.

Page 5, lines 20-24: More explanations are needed.

Page 5, line 31: If I understand it correctly, the currents are of opposite directions in the layers above sigma-theta 14.5 and below it. If it is true, it requires an extended discussion on the influence of this feature on the oxygen distribution.

Page 6, lines 8-17: This paragraph is very hard to understand for English. For example, I believe the first sentence should be "the oxygen distribution versus depth is exceptionally dynamic". The other sentences are equally vague in the present form.

Page 6, lines 23-26: This justification for hypoxia should be somewhere in Introduction
section.

Page 8, lines 5-8: I do not see any reason for this paragraph here.

Page 9, References: I believe there are too many references for publications by the authors of this manuscript. There usually should be up to 25% of them, but not more.

Figure 1. This figure is mentioned once and has never been discussed. Consider deleting it.

Figure 3. This figure is very complicated and hard to understand or follow.

Figure 5. This figure is mentioned once and only for the sharp oxycline. I do not see any reason to plot several profiles and to apply 20 uM shift.

---

## Referee Comment (RC2) · Anonymous Referee #2 · 24 Sep 2018

Ostrovskii et al use a new, high-resolution, multi-variable dataset from a moored profiler in the slope waters of the NE Black Sea to study oxygen dynamics in short time scales (hours to weeks). They find that the hypoxic boundary fluctuated mostly with two time scales: the inertial period ($\sim$17 hs) and the passage of current meanders and eddies ($\sim$5 days). The work is novel mostly due to being the first study in the region showing results from such a highly resolved dataset and the topic is quite interesting and relevant for all oxygen-inclined ocean scientists. However, the manuscript requires significant improvements. I find that several conclusions or statements made by the authors need further explanations/support. Also, and as mentioned by another reviewer (Sergey Konovalov), both the organization and the use of English language of the paper need to be improved. While I give some suggestions in my comments,

the writing requires an in-depth revision beyond those suggestions; I feel many sentences/paragraphs need to be completely rewritten to improve the readability and the flow of the text. Overall, I recommend the publication of this work after major revisions are done to address the above issues.

General comments:

Abstract: it needs to be streamlined to better convey the importance of the work and its results (there are several good resources online on how to write a good scientific abstract). For instance, it should start with the big picture – the posing of the problem you are studying and its importance (rather than mentioning the importance of the methods) – and end with the main conclusions and broader implications. Furthermore, in its current form, the abstract has too many details about the sensors used. In the abstract and the main text: authors say that their study looks at time scales from hours to months. However, they have data for two months, so it would be more appropriate to say from "hours to weeks" instead.

Introduction: The first few paragraphs, where the Black Sea is described, should also include a thorough description of the horizontal and vertical structure. That information is needed later on, and sometimes it is scattered in the text or assumed to be known by the reader. In addition, all the terminology needs to be properly defined in this section (e.g. suboxic layer, hydrogen sulfide boundary, oxygen penetration layer, hypoxia onset) and used consistently in the rest of the paper.

Results, Discussion: there are some results in the Discussion section and some discussion in the Results section. Make sure each section contains only results and only discussion, or consider merging both into a "Results and Discussion" section.

Everywhere: There are many paragraphs that are one- or two-sentence long, which should be avoided. Furthermore, many citations are within parenthesis, when they should read instead "Lastname et al. (year)".

Figures: some captions tend to have a lot of information that should be presented instead (or as well) in the main text.

Specific comments:

Page2.Line1: "resistance": probably the word you are looking for is resilience? Anyways, the sentence does not read well and should be rewritten.

P2.L8-9: "This memory affects the distribution. . .": you should explain better why phytoplankton is sensitive to the properties below the seasonal thermocline. The next sentence explains it (it is a source of nutrients), but the sentence is presented as a separate concept (by starting with "In addition").

P2.L21: suboxic layer needs to be defined.

P3.L6: hydrogen sulfide boundary has to be defined.

P3.L10-21: this paragraph is a weird way of posing the rationale for the setup of the mooring. It needs to be rewritten and bullets should be removed.

P3.L26: I don't think that "relief" is the word you are looking for. Topography? Roughness?

P3.L29: "on" the shelf or "over" the shelf

P3.L30-33: this is an example of a sentence that is too long. Aim for sentences up to 2 lines-long

P4.8-9: unless some description is added for each section, this sentence should be removed.

P4.10: Odd heading. Maybe just "Methods" or "Observations"?

P4.17: if profiles occur every 2 hours, how can you obtain 24 profiles in a day? Did you mean 12 profiles per day? Otherwise, please explain better

P5.17: "seasonal": talking about seasonality with two months of data is a bit of a

stretch. Remove "seasonal". Furthermore, the trend is hard to see in figure 2, maybe a different color scale would help.

P5.18: refer to Fig 3 at the end of the sentence

P5.rest of the page: the authors start with the conclusion (temporal variability linked to meandering jet and eddies, line 19-20) but present some kind of evidence later on (lines 28-30). The logic and flow here need to be streamlined.

P6.L5 and rest of the manuscript: instead of a long dash "–" we see a "Ãů"

P6.L6-7: Please explain more explicitly how you get to the Itotal number of 33 – 34 mol m-2

P6.L12-15: Please explain here the chosen vertical axes for figures 6 and 7, otherwise the reader starts to wonder why the authors did not choose just depth. Also, introduce in the main text the same notation as used in the figures and their caption (or remove said notation from the figures).

P6.L17: shouldn't Fig 2 be cited at the end of the sentence?

P6.L18: About the 89%: there is information explaining this number in the caption of figure 7 that should be moved here.

P6.L22: suggest removing seasonal or replacing by something more appropriate.

P6.L23: remove "follows:"

P6.L29-30: belongs to Discussion rather than Results

P6.L30: "as a result of horizontal advection"→ this statement seems like a reasonable assumption, but it should be better justified. Or even better, would it be possible to prove it using your velocity data and/or knowledge of surrounding water masses?

P7.L8: replace "is" by "was"

P7.L26: remove "layer of"

P7.L30-34: belongs to Results rather than Discussion

P8.L1-8: How do the studies mentioned here relate to this work?

P8.L11-12: merge both paragraphs.

P8.L29-33: some of these summarized results were not mentioned in the Results section (e.g. "11% of profiles" part – I found it in Fig 6 caption, but not in the main text). In addition, why is the winter of 2016 warm? The "warm winter" statement in line 20 should be described early on (in the Results if it can be proved with the profiler data or in the Introduction if other references are used to describe it) – note that it should be proven specifically for 2016, not just generically claiming that global warming has warmed the Black Sea.

P9.L8-9: "water in this new CIL is rather warm" – I find this confusing, because earlier you mentioned that it was colder than the old CIL. Is the new CIL warm or cold?

P9.L10: what essential changes in plankton migration may occur? Please expand

P9.L19: Show Kerch Strait on map in fig 1.

Fig 1: As mentioned above, show Kerch Strait if you are mentioning in the main text. It would be beneficial to have an inset showing the whole Black Sea and a little box showing the location of the map in the figure.

Fig 3: along –shore axis is 50deg and cross-shore axis 310deg → why are they not perpendicular? (Also, this information belongs also to the main text)

Fig 4, 8: If possible, always try to choose colors that are color-blind safe (i.e., avoid red and green in the same figure).

Fig 6, 7, 10: I already mentioned the Y axes and that its description should be properly given in the main text. Also, there is info if Fig 6 that belongs to the main text.

---

## Referee Comment (RC3) · E. Stanev (Referee) · 1 Oct 2018

This is a clear study presenting an interesting observational material and providing interpretations of findings. Results are convincing, however the presentation needs some improvements. I hope that below comments can help authors provide improved revision.

1. English grammar needs substantial improvement by native English speaking oceanographer. 2. Explain more clearly concepts and literature review (see further some specific comments). 3. I do not find a clear prove that "the new CIL emerged by horizontal advection above the pycnocline only at the end of the observational survey". This is mentioned in the abstract and in p. 5, 6 where authors mention Fig. 9. They

should explain how they deduce this from the figure. 4. When talking about time scales authors could make spectral analysis to quantify the relevant periods: "the hypoxia boundary depth fluctuated on two time scales: ∼17 h due to the inertial oscillations and approximately 5 days due to the current meanders and eddies. 5. Stanev et al., 1995 is cited but not added in the reference list. 6. When you say "In general, factors that provide the resistance of the Black Sea ecosystem to anthropogenic and climatic effects are weaker than those in other marginal seas adjacent to the European continent" cite the origin of this statement or provide evidence. 7. The statement "In the wintertime, intensive cooling and vertical convective mixing are known to occur in the Black Sea, which enables the top layer to achieve its maximum thickness and minimum temperature (refer to Piotukh et al., 2011 and corresponding references)." is too general. It is not Piotukh et all. (2011), who first came to this conclusion. Authors should cite relevant papers. Perhaps stress that Piotukh et all. (2011) came to this conclusion for areas subject to the present research. When describing the formation of CIL take in discussion Stanev, E. V., M. J. Bowman, E. L. Peneva, and J. V. Staneva (2003b) Control of Black Sea intermediate water mass formation by dynamics and topography: comparisons of numerical simulations, survey and satellite data. J. Mar. Res., 61, 59-99. 8. Rephrase "However, the ARGO buoys profiled the water column at five-day intervals. Therefore, their data could not be used to evaluate typical time scales of short-time fluctuations in the oxygen inventory.". Some Argo floats in the Black Sea are programmed to measure with finer sampling rate. Re-programming can be done during the operations. 9. The phrase "In the center of the western cyclonic gyre, the depth of the top mixed layer was limited by 40 m also in an anomalously cold winter (Gregg, Yakushev, 2005)." has to be specified in the context of present paper. Is what happens in the western gyre relevant to the eastern part, the latter being the subject of this study. If yes, prove it. 10. p. 8, l. 5, In the paper of Gregg and Yakushev there is no a word "hypoxia". They talk about SOL. To my knowledge suboxia is more used when describing shelf processes. Please check with hydrochemisists. 11. Authors could consider studying the relevance of following papers Stanev, E. V., S. Grayek,

H.Claustre, C. Schmechtig, and A. Poteau (2017) Water intrusions and particle signatures in the Black Sea: a Biogeochemical-Argo float investigation. Ocean Dynamics, 67, 1119–1136, doi: 10.1007/s10236-017-1077-9 and Stanev, E. V., Poulain, P.-M., Grayek, S., Johnson, K. S., Claustre, H., & Murray, J. W. (2018). Understanding the dynamics of the oxic-anoxic interface in the Black Sea. Geophysical Research Letters, 45. https://doi.org/10.1002/2017GL076206 to their study.

Emil Stanev

---

## Referee Comment (RC4) · Anonymous Referee #4 · 7 Oct 2018

General Comments ******************** In this manuscript the authors present and analyze a extremely valuable dataset gathered from a moored profiler equipped with Doppler and oxygen sensors on the edge of the Black Sea North Eastern shelf. Based on this datasets, they address the issue of the short time-scales of variations for vertical horizons characterizing the density and oxygenation structure. Several recent publication have shown the timeliness of this study, in the context of Black Sea deoxygenation, as such detailed experimental data set provides the means to evidence diapycnal ventilation processes and to quantify the variability of bottom oxygen conditions on the narrow eastern shelf. As discussed extensively by other reviewers, the manuscript would gain a lot by revising the structure and language, and therefore requires substantial revision. However, we encourage the authors in tackling this effort since the materials

and results absolutely deserves publication. Here follow some additional suggestions, and comments.

⇢ A major object of the manuscript (and a strong asset of the sampling approach) regards the characterization of the temporal scales of vertical oscillations. In particular, it is very interesting to perceive phenomena acting at time scales shorter than those typical of current sampling in the open sea sampling frequency (eg. Argo). I would suggest, as I think this is accessible , to analyze those time scales more robustly with spectral analysis tools, applied, for instance on the time series of SOL, onset depth of hypoxia, velocity components a certain depth, etc.. This could feed the discussion by putting processes in relation with (ranges of) time scales more clearly.

⇢ The discussion on oxygen inventory is somewhat confusing, since it may give, at a fast read, the impression that the observed variations in oxygen inventory are to be related to oxygen sinks/sources terms. It later appears that the variations in this oxygen inventory (integrated over depth) is mostly due to widening/thickening of isopycnals intervals, meaning that oxygen waters enter and leaves this particular water column laterally as waves displaces the isopycnal surfaces vertically. I believe this should be clarified earlier. Maybe (suggestion) the wording "local oxygen inventory" may serve this clarification.

⇢ Reporting on velocities : 1) I think there is a confusion in the directions given as referential for the velocities. 1a) cross and along-shore directions are switched in legend of Fig 3. 1b) the two direction given are not perpendicular ? 2) Could the authors consider the option to report velocities with intensity (magnitude) on one panel, and direction on a second panel (using a cyclic color scale).

Specific comments ******************** ⇢ P1L9 : How the fact that the benthic habitat is small compared to the tota Black Sea area does aggravate the stress of the ecosystem ? Could you clarify this statement ? ⇢ P2L1-2 replace "resistance" with "resilience". This sentence should be detailed and referenced. ⇢ P2L3-26 Please rephrase this

paragraph, the reading is somewhat unclear. (eg. 'downwelling in the coastal part and upwelling in the central part', or refer to curvarture of isopycnal surfaces) • P3L25 the term "minimum" is unclear here. • P3L28 "Wind-induced upwellings, although .." ( add "-" and "," ) • P4L13 ".. deployed at a depth of approximately .." ( add "a depth of") • P4L22 "smoothly worked" -> "worked smoothly" • P5L26 week -> weak • P7L24-26 -> a part of this sentence should go to methods • Fig 6,7 I suggest to define the vertical referential used here in the methods (for instance referring to "normalized depth" or some similar wordings) , and to use this definition in the figures and captions. In addition, I'd indicate an horizontal mark at the mean depth of maximum density gradient, that is used as a reference. On the same topic: was a shift of vertical coordinates also operated for fig 11? if yes, please indicate so, if no, could you explain why ? • Fig 10: To support the discussion on this figure, it would be useful to get a similar picture for the cases Is14 > 3.5.

References *********** P1L25 Reference Stanev et al 1995 does not appear in the bibliography. P1L28 Capet et al 2004 does not appear in the bibliography. P1L28 Capet et al 2014 does not appear in the bibliography. P6L24 Middelburg & Levin 2009 does not appear in the bibliography. P7L19 Zatsepin et al 2011 does not appear in the bibliography. General : add "and" when referencing two-authors references.

---

## Author Comment (AC1) · 19 Nov 2018

We thank the reviewer for the suggestions that helped to improve the manuscript. In the revised manuscript, we try to reform its structure. In particular, we introduced two subsections in the section Results, which aim at more clear presentation of the main findings such as the periodic inertial oscillations in the distribution of oxygen. We also submitted the manuscript for advanced editing at Wiley Editing Services that is among the best language services, as far as we know.

Our response to the reviews includes: (1) comments from referees and the author's response, (2) the author's changes in manuscript (a marked-up manuscript version), (3) the revised version of the manuscript.

[Figure]

Here is our response to the comments of the review #1.

Consider to change the title for "The oxygen dynamics at the shelf edge of the Black Sea on the timescale of hours-to-days". Our respond: The title of the ms was modified to emphasize the timescales of the oxygen variability.

Consider to delete the last sentence in the abstract because it is hardly the major and final result of this work. Our respond: The last sentence in the abstract was deleted.

Page 1, line 15: Unless I have missed it, any explanation or justification of the chosen depth of 30 meters has not been suggested in the text. Our respond: The profiler Aqua-log operates below the subsurface floatation. Another instrument would be needed for observations in the top sea layer near the air-sea interface. We worked on such instrumentation recently and tested it in the Black Sea last month. Anyway, the explanation you requested is added into the section Methods.

Page 2, lines 1-2: This sentence comes from nowhere. Suggest a reference or some justification. Our respond: The reference is added.

Page 2, lines 5-6: CIL and other specific for the Black Sea features need an explanation or references. Our respond: The CIL and other important features of the stratification are defined in Introduction.

Page 2, line 11: The referenced publication by Oguz et al. is good but it is about modeling. Look for the recent publication by Kubryakov and Stanichniy or recently published data by Yunev et al., Mikaelyan et al. based on observations. Our respond: We decided to streamline the text as the second reviewer suggested. Hence we omitted these lines.

Page 3, line 5: I do not see any reason for discussions of global warming in this manuscript. Our respond: We want to mention a general context for the warming of the CIL.

Page 5, lines 20-24: More explanations are needed. Our respond: We added the
satellite image showing the mesoscale and submesoscale eddies in one of the days during the survey.

Page 5, line 31: If I understand it correctly, the currents are of opposite directions in the layers above sigma-theta 14.5 and below it. If it is true, it requires an extended discussion on the influence of this feature on the oxygen distribution. Our respond: This issue certainly needs special attention. But it is beyond the scope of our ms. We hope to publish relevant findings elsewhere.

Page 6, lines 8-17: This paragraph is very hard to understand for English. For example, I believe the first sentence should be "the oxygen distribution versus depth is exceptionally dynamic". The other sentences are equally vague in the present form. Our respond: The text is edited.

Page 6, lines 23-26: This justification for hypoxia should be somewhere in Introduction section. Our respond: Done.

Page 8, lines 5-8: I do not see any reason for this paragraph here. The lines 1-8 are deleted.

Page 9, References: I believe there are too many references for publications by the authors of this manuscript. There usually should be up to 25% of them, but not more. Our respond: We added several new references so the number of self-citations falls below 25%.

Figure 1. This figure is mentioned once and has never been discussed. Consider deleting it. Our respond: We modified the figure and mentioned it several times in the revised ms.

Figure 3. This figure is very complicated and hard to understand or follow. Our respond: This figure shows complexity of the current structure over the continental slope. The upper figure shows the along-shore current profile vs time, the lower figure shows the cross-shore current also in the depth-time plane. The positive directions are Northwestward and Northeastward (shown in red). The reversal currents are shown in blue. The isopycnals are superimposed. The figure clearly shows that the currents and the density variations and therefore the oxygen dynamics are coherent.

Figure 5. This figure is mentioned once and only for the sharp oxycline. I do not see any reason to plot several profiles and to apply 20 uM shift. Our respond: We deleted 3 of 5 profiles and showed the rest 2 profiles without the shift in the revised figure.  

Please also note the supplement to this comment:
https://www.ocean-sci-discuss.net/os-2018-91/os-2018-91-AC1-supplement.pdf

---

## Author Comment (AC2) · 19 Nov 2018

Dear Reviewer:

We thank you for many judicious comments and helpful suggestions. Certain parts of the manuscript were rewritten and restructured. We went through the comments of the other three referees and made many other modifications in the paper. In particular, we applied the spectral analysis to justify the inertial variations of the oxygen stratification. The references were expanded. The figures and the figure captions were updated to comply with the requests of the referees. We decided to show the satellite imagery that indicates typical space structures over the study area that is important for identifying the dynamical processes at ocean mesoscale and submesoscale that are responsible

for oxygen dynamics at timescales of several days. Our fellow researcher, Dr. Dmitry Soloviev, processed the satellite imagery and we invited him to be a co-author of this manuscript. The manuscript title was slightly changed to be more specific: The Short Timescale Variability of the Oxygen Inventory in the NE Black Sea Slope Water. Finally the manuscript was submitted to Wiley Editing Service for advanced editing. Many edits were made to further improve the flow and readability of the text. Some sentences were restructured to address the overly complex structure or to revise potentially unclear phrasing. In cases where the meaning of the text was not clear, revisions were made to convey the information with increased clarity and reduced ambiguity. Some edits were made to improve conciseness by trimming unnecessary words and streamlining the flow of the manuscript. The manuscript was edited by Wiley Editing Service for grammar, phrasing, and punctuation. This our response to your review (see supplement, please) includes: (1) the reviewer's comments and the author's response, (2) the author's changes in manuscript (a marked-up manuscript version), (3) the revised version of the manuscript.

On behalf of the authors, Alexander Ostrovskii

Shirshov Institute of Oceanology Russian Academy of Sciences Nahimovskiy prospect, 36 Moscow 117997 E-mail: osasha@ocean.ru Phone: +7-916-4905969

Please also note the supplement to this comment:
https://www.ocean-sci-discuss.net/os-2018-91/os-2018-91-AC2-supplement.pdf

**Supplement:**

**Response to Review #2**

Dear Reviewer:

We thank you for many judicious comments and helpful suggestions. Certain parts of the manuscript were rewritten and restructured.

We went through the comments of the other three referees and made many other modifications in the paper. In particular, we applied the spectral analysis to justify the inertial variations of the oxygen stratification. The references were expanded. The figures and the figure captions were updated to comply with the requests of the referees. We decided to show the satellite imagery that indicates typical space structures over the study area that is important for identifying the dynamical processes at ocean mesoscale and submesoscale that are responsible for oxygen dynamics at timescales of several days. Our fellow researcher, Dr. Dmitry Soloviev, processed the satellite imagery and we invited him to be a co-author of this manuscript.

The manuscript title was slightly changed to be more specific: The Short Timescale Variability of the Oxygen Inventory in the NE Black Sea Slope Water.

Finally the manuscript was submitted to Wiley Editing Service for advanced editing.

Many edits were made to further improve the flow and readability of the text. Some sentences were restructured to address the overly complex structure or to revise potentially unclear phrasing. In cases where the meaning of the text was not clear, revisions were made to convey the information with increased clarity and reduced ambiguity. Some edits were made to improve conciseness by trimming unnecessary words and streamlining the flow of the manuscript. The manuscript was edited by Wiley Editing Service for grammar, phrasing, and punctuation.

This our response to your review includes: (1) the reviewer's comments and the author's response, (2) the author's changes in manuscript (a marked-up manuscript version), (3) the revised version of the manuscript.

On behalf of the authors,
Alexander Ostrovskii

Shirshov Institute of Oceanology
Russian Academy of Sciences
Nahimovskiy prospect, 36
Moscow 117997
E-mail: osasha@ocean.ru
Phone: +7-916-4905969

**Here is our response (in bold font) to the comments of the review #2.**

1.      Abstract: it needs to be streamlined to better convey the importance of the work and its results (there are several good resources online on how to write a good scientific abstract). For instance, it should start with the big picture – the posing of the problem you are studying and its importance (rather than mentioning the importance of the methods) – and end with the main conclusions and broader implications. Furthermore, in its current form, the abstract has too many details about the sensors used. In the abstract and the main text: authors say that their study looks at time scales from hours to months. However, they have data for two months, so it would be more appropriate to say "from hours to weeks" instead.

**The abstract was modified. In particular, the details about the sensors are omitted. The definition of the short timescale as "from hours to weeks" is used throughout the revised ms.**

2.      Introduction: The first few paragraphs, where the Black Sea is described, should also include a thorough description of the horizontal and vertical structure. That information is needed later on, and sometimes it is scattered in the text or assumed to be known by the reader. In addition, all the terminology needs to be properly defined in this section (e.g. suboxic layer, hydrogen sulfide boundary, oxygen penetration layer, hypoxia onset) and used consistently in the rest of the paper.

**The section Introduction was rewritten to address the reviewer concern. The definitions of important features of the oxygen stratification as well as the Cold Intermediate Layer are presented.**

3.      Results, Discussion: there are some results in the Discussion section and some discussion in the Results section. Make sure each section contains only results and only discussion, or consider merging both into a "Results and Discussion" section

**The material of these sections is rearranged (see also below). The revised section Results has 2 subsections 3.1 Background observations and 3.2 Oxygen dynamics**

4.      Everywhere: There are many paragraphs that are one- or two-sentence long, which should be avoided. Furthermore, many citations are within parenthesis, when they should read instead "Lastname et al. (year)"
**The Copernicus Publications Word template is used to prepare the ms including the citations.**

5.      Figures: some captions tend to have a lot of information that should be presented instead (or as well) in the main text.
**Information of certain figure captions is also presented in the revised ms.**

6.      Specific comments:

Page2.Line1: "resistance": probably the word you are looking for is resilience? Anyways, the sentence does not read well and should be rewritten.
**Corrected**

P2.L8-9: "This memory affects the distribution…": you should explain better why phytoplankton is sensitive to the properties below the seasonal thermocline. The next sentence explains it (it is a source of nutrients), but the sentence is presented as a separate concept (by starting with "In addition").
**Corrected**

P2.L21: suboxic layer needs to be defined.
**Defined**

P3.L6: hydrogen sulfide boundary has to be defined
**Defined**

P3.L10-21: this paragraph is a weird way of posing the rationale for the setup of the mooring. It needs to be rewritten and bullets should be removed.
**Rewritten**

P3.L26: I don't think that "relief" is the word you are looking for. Topography? Roughness?
**Rewritten**

P3.L29: "on" the shelf or "over" the shelf
**Corrected**

P3.L30-33: this is an example of a sentence that is too long. Aim for sentences up to 2 lines-long

P4.8-9: unless some description is added for each section, this sentence should be removed.
**Removed**

P4.10: Odd heading. Maybe just "Methods" or "Observations"?
**The heading "Methods" is used.**

P4.17: if profiles occur every 2 hours, how can you obtain 24 profiles in a day? Did you mean 12 profiles per day? Otherwise, please explain better
**The explanation is given.**

P5.17: "seasonal": talking about seasonality with two months of data is a bit of a stretch. Remove "seasonal". Furthermore, the trend is hard to see in figure 2, maybe a different color scale would help.
**This sentence is rewritten.**

P5.18: refer to Fig 3 at the end of the sentence
**Done**

P5.rest of the page: the authors start with the conclusion (temporal variability linked to meandering jet and eddies, line 19-20) but present some kind of evidence later on (lines 28-30). The logic and flow here need to be streamlined.
**The material of the Section is rearranged to address this point.**

P6.L5 and rest of the manuscript: instead of a long dash "–" we see a "Ã˚u"
**Corrected**

P6.L6-7: Please explain more explicitly how you get to the Itotal number of 33 – 34 mol m-2
**Corrected**

P6.L12-15: Please explain here the chosen vertical axes for figures 6 and 7, otherwise the reader starts to wonder why the authors did not choose just depth. Also, introduce in the main text the same notation as used in the figures and their caption (or remove said notation from the figures).

**The explanation is given at the end of Section 2.**

P6.L17: shouldn't Fig 2 be cited at the end of the sentence?
**The figure is cited here.**

P6.L18: About the 89%: there is information explaining this number in the caption of figure 7 that should be moved here.
**This information is moved to the main text.**

P6.L22: suggest removing seasonal or replacing by something more appropriate
**"Seasonal" is deleted.**

P6.L23: remove "follows:"
**"Follows:" deleted. The definition of hypoxia is moved to Introduction.**

P6.L29-30: belongs to Discussion rather than Results
**That sentence was omitted.**

P6.L30: "as a result of horizontal advection"! this statement seems like a reasonable assumption, but it should be better justified. Or even better, would it be possible to prove it using your velocity data and/or knowledge of surrounding water masses?
**We are working on justification of the cold intermediate water advection by using the profiler data since 2012. The analysis would take more time and the results will be published later on.**

P7.L8: replace "is" by "was"
**Done.**

P7.L26: remove "layer of"
**Done.**

P7.L30-34: belongs to Results rather than Discussion
**This paragraph is moved to Results.**

P8.L1-8: How do the studies mentioned here relate to this work?
**These lines are deleted.**

P8.L11-12: merge both paragraphs.

**The text is modified.**

P8.L29-33: some of these summarized results were not mentioned in the Results section (e.g. "11% of profiles" part – I found it in Fig 6 caption, but not in the main text).
In addition, why is the winter of 2016 warm? The "warm winter" statement in line 20 should be described early on (in the Results if it can be proved with the profiler data or in the Introduction if other references are used to describe it) – note that it should be proven specifically for 2016, not just generically claiming that global warming has warmed the Black Sea

**Data of the weather station on the air temperature are added into the section 3.1.**

P9.L8-9: "water in this new CIL is rather warm" – I find this confusing, because earlier you mentioned that it was colder than the old CIL. Is the new CIL warm or cold?

**Corrected.**

P9.L10: what essential changes in plankton migration may occur? Please expand

**Done**

P9.L19: Show Kerch Strait on map in fig 1

**Done**

Fig 1: As mentioned above, show Kerch Strait if you are mentioning in the main text.
It would be beneficial to have an inset showing the whole Black Sea and a little box showing the location of the map in the figure.

**Done**

Fig 3: along –shore axis is 50deg and cross-shore axis 310deg ! why are they not perpendicular? (Also, this information belongs also to the main text)

**The mistake is corrected.**

Fig 4, 8: If possible, always try to choose colors that are color-blind safe (i.e., avoid red and green in the same figure).

**We changed the colors at certain figures in some cases we could not change the colors because it made the figures unclear.**

Fig 6, 7, 10: I already mentioned the Y axes and that its description should be properly given in the main text. Also, there is info if Fig 6 that belongs to the main text. **Modified.**

[revised manuscript text omitted]

---

## Author Comment (AC3) · 19 Nov 2018

Dear Reviewer:

Thank you very much for careful reading of our manuscript. We revised it in line with your comments. Below we list your comments together with our responses.

1. English grammar needs substantial improvement by native English speaking oceanographer. Our respond: The paper was rewritten and polished by the Wiley Language Service.

2. Explain more clearly concepts and literature review (see further some specific comments). Our respond: The definitions of the important features of the sea stratification were put into Introduction. The literature cited was expanded.

3. I do not find a clear prove that "the new CIL emerged by horizontal advection above the pycnocline only at the end of the observational survey". This is mentioned in the abstract and in p. 5, 6 where authors mention Fig. 9. They should explain how they deduce this from the figure. Our respond: At the end of the survey, the temperature profile showed two minimums. The lower one was located at the same depth range below the pycnocline core where the temperature was above 8.55C. The upper colder minimum emerged on March 3 above the pycnocline core.

4. When talking about time scales authors could make spectral analysis to quantify the relevant periods: "the hypoxia boundary depth fluctuated on two time scales: âLij17 h due to the inertial oscillations and approximately 5 days due to the current meanders and eddies. Our respond: The spectral analysis is carried out for the data ensemble of the hypoxia onset depth of January 2016 when there was no gap in the data. The spectrum is plotted and inserted in Results as a new figure.

5. Stanev et al., 1995 is cited but not added in the reference list. Our respond: The missing reference is added.

6. When you say "In general, factors that provide the resistance of the Black Sea ecosystem to anthropogenic and climatic effects are weaker than those in other marginal seas adjacent to the European continent" cite the origin of this statement or provide evidence. Our respond: The reference to the paper by A. E. Kideys (Fall and rise of the Black Sea ecosystem. Science, 297, 1482-1484, doi:10.1126/science.1073002, 2002) is added.

7. The statement "In the wintertime, intensive cooling and vertical convective mixing are known to occur in the Black Sea, which enables the top layer to achieve its maximum thickness and minimum temperature (refer to Piotukh et al., 2011 and corresponding references)." is too general. It is not Piotukh et all. (2011), who first came to this conclusion. Authors should cite relevant papers. Perhaps stress that Piotukh et all. (2011) came to this conclusion or areas subject to the present research. When OSD
describing the formation of CIL take in discussion Stanev, E. V., M. J. Bowman, E. L. Peneva, and J. V. Staneva (2003b) Control of Black Sea intermediate water mass formation by dynamics and topography: comparisons of numerical simulations, survey and satellite data. J. Mar. Res., 61, 59-99. Our respond: Proper references are added

8. Rephrase "However, the ARGO buoys profiled the water column at five-day intervals. Therefore, their data could not be used to evaluate typical time scales of short-time fluctuations in the oxygen inventory.". Some Argo floats in the Black Sea are programmed to measure with finer sampling rate. Re-programming can be done during the operations. Our respond: The text was modified to take this into account.

9. The phrase "In the center of the western cyclonic gyre, the depth of the top mixed layer was limited by 40 m also in an anomalously cold winter (Gregg, Yakushev, 2005)." has to be specified in the context of present paper. Is what happens in the western gyre relevant to the eastern part, the latter being the subject of this study. If yes, prove it. Our respond: These lines were deleted.

10. p. 8, I. 5, In the paper of Gregg and Yakushev there is no a word "hypoxia". They talk about SOL. To my knowledge suboxia is more used when describing shelf processes. Please check with hydrochemisists. Our respond: These lines were deleted.

11. Authors could consider studying the relevance of following papers Stanev, E. V., S. Grayek, H.Claustre, C. Schmechtig, and A. Poteau (2017) Water intrusions and particle signatures in the Black Sea: a Biogeochemical-Argo float investigation. Ocean Dynamics, 67, 1119–1136, doi: 10.1007/s10236-017-1077-9 Stanev, E. V., Poulain, P.-M., Grayek, S., Johnson, K. S., Claustre, H., & Murray, J. W. (2018). Understanding the dynamics of the oxic-anoxic interface in the Black Sea. Geophysical Research Letters, 45. https://doi.org/10.1002/2017GL076206 to their study Our respond: The reference was added to the ms.

Sincerely yours, Alexander Ostrovskii

OSD
Please also note the supplement to this comment: https://www.ocean-sci-discuss.net/os-2018-91/os-2018-91-AC3-supplement.pdf

---

## Author Comment (AC4)

**Response to review #4**

We are very grateful to the Reviewer for a very careful reading of the manuscript and useful comments and suggestions. Below we list the comments together with our responses to them (in bold font).

**General Comments**

5

In this manuscript the authors present and analyze a extremely valuable dataset gathered from a moored profiler equipped with Doppler and oxygen sensors on the edge of the Black Sea North Eastern shelf.

- 10 Based on this datasets, they address the issue of the short time-scales of variations for vertical horizons characterizing the density and oxygenation structure. Several recent publication have shown the timeliness of this study, in the context of Black Sea deoxygenation, as such detailed experimental data set provides the means to evidence diapycnal ventilation processes and to quantify the variability of bottom oxygen conditions on the narrow eastern shelf. As discussed extensively by other reviewers, the
- 15 manuscript would gain a lot by revising the structure and language, and therefore requires substantial revision. However, we encourage the authors in tackling this effort since the materials and results absolutely deserves publication. Here follow some additional suggestions, and comments.
- 20 1. A major object of the manuscript (and a strong asset of the sampling approach) ' regards the characterization of the temporal scales of vertical oscillations. In particular, it is very interesting to perceive phenomena acting at time scales shorter than those typical of current sampling in the open sea sampling frequency (eg. Argo). I would suggest, as I think this is accessible , to analyze those time scales more robustly with
- 25 spectral analysis tools, applied, for instance on the time series of SOL, onset depth of hypoxia, velocity components a certain depth, etc.. This could feed the discussion by putting processes in relation with (ranges of) time scales more clearly.

The inertial scale oscillations of the hypoxia onset depth were verified by using D. Thomson's multitaper method that became a conventional tool for time series analysis in Matlab. When used

- 30 with Matlab's sptool the MTM also delivers the confidence intervals if the power spectrum estimate. In the revised ms, the spectral analysis is carried out for the data ensemble of the hypoxia onset depth of January 2016 when there was no gap in the data. The spectrum is plotted and inserted in Results as a new figure.
- 35 2. The discussion on oxygen inventory is somewhat confusing, since it may give, at ' a fast read, the impression that the observed variations in oxygen inventory are to be

related to oxygen sinks/sources terms. It later appears that the variations in this oxygen inventory (integrated over depth) is mostly due to widening/thickening of isopycnals intervals, meaning that oxygen waters enter and leaves this particular water column laterally as waves displaces the isopycnal surfaces vertically. I believe this should be

5 clarified earlier. Maybe (suggestion) the wording "local oxygen inventory" may serve this clarification.

We agree about the term "local" and inserted it at proper parts of the text. If our next research grant would be approved by Russian Fund for Basic Research, we hope to deploy two profiler moorings to get the data about the horizontal advection of dissolved oxygen.

- 10 Perhaps, under international cooperation we can deploy at least 3 profilers to approach this problem more comprehensively.
  - 3. Reporting on velocities :
  - 1) I think there is a confusion in the directions given '
- as referential for the velocities. 1a) cross and along-shore directions are switched in legend of Fig 3. 1b) the two direction given are not perpendicular ?
   This mistake is corrected.

2) Could the authors consider the option to report velocities with intensity (magnitude) on one panel,

20 and direction on a second panel (using a cyclic color scale).

We wanted to focus on the oxygen dynamics rather than the current variability although the latter is very interesting on its own right. The along-shore current component was much stronger that the cross-shore component. This could be shown by using angle histogram. Actually, we are analyzing the current meter data at companion moorings also operated in 2016, which

25 unfortunately were not equipped with the oxygen sensors. The results will be published elsewhere.

Specific comments

P1L9 : How the fact that the benthic habitat is small compared to the total Black Sea area does

30 aggravate the stress of the ecosystem? Could you clarify this statement ?

**The reference to the paper by A. E. Kideys (Fall and rise of the Black Sea ecosystem. Science, 297, 1482-1484, doi:10.1126/science.1073002, 2002) is added.**

P2L1-2 replace "resistance" with "resilience". '

35 This sentence should be detailed and referenced.The "resistance" is substituted for "resilience".

P2L3-26 Please rephrase this paragraph, the reading is somewhat unclear. (eg. 'downwelling in the coastal part and pwelling in the central part', or refer to curvarture of isopycnal surfaces) **The style is imporved.**

5 P3L25 the term "minimum" is unclear here. "minimum" is replaced by "more than"

P3L28 "Wind-induced upwellings, although .." ( add "-" and "," ) **The style is imporved.**

10

P4L13 ".. deployed at a depth of approximately .." ( add "a depth of") **The style is imporved.**

P4L22 "smoothly worked" -> "worked smoothly"

15 P5L26 week -> weak

**Corrected**

P7L24-26 -> a part of this sentence should go to methods

**20 **Done**

Fig 6,7 I suggest to define the vertical referential used here in the methods (for instance referring to "normalized depth" or some similar wordings), and to use this definition in the figures and captions. In addition, I'd indicate an horizontal mark at the mean depth of maximum

25 density gradient, that is used as a reference. On the same topic: was a shift of vertical coordinates also operated for fig 11? if yes, please indicate so, if no, could you explain why ?

The estimates of the mean depth of the maximum density gradient re inserted in the ms (P.7 L.17, 26)

**30 In Fig. 11, the Richardson numbers are plotted vs depth.**

Fig 10: To support the discussion on this figure, it would be useful to get a ' similar picture for the cases Is14 > 3.5.

**The figure is added.**

[revised manuscript text omitted]

---

## Author Response (AR2)

December 7, 2018

Piers Chapman
Editor
Ocean Science
An interactive open-access journal of the European Geosciences Union

**Re:     Response to Topic Editor Comments to the Author of the Ocean Science paper os-2018-91**

Dear Prof. Chapman,

We really appreciate your comments that helped to improve our paper.  The Ocean Science's openness makes this journal very attractive for authors. Below is a list of relevant changes made in the paper after your final decision:

>I did, however, note one or two minor errors/omissions:
>1.     There are several references to a paper by Yakushev et al (2005), but I couldn't find this >in the reference list.

The reference was inserted.

>2.     Ostrovskii and Zatsepin (2011) is listed in the reference list, but does not  appear in the >text.

The reference was omitted.

>3.     I think the caption to Fig 3 is still incorrect. Given the shape of the coastline, >shouldn't > the alongshore axis be orientated at 320°, and the cross-shelf axis at  50°?

I mistakenly swapped the directions in the figure caption. The correction was made.

>4.     The caption for Fig 6 needs correcting now that it shows only two profiles, and not five. >It should read: "Examples of two profiles[O2] j (z)…in the top left >corner. Profile 145 was >taken when the probe was moving up, and profile 146 when the probe was moving down…."

The figure caption was corrected.

Thank you very much again for your efforts.

Sincerely yours,
Alexander Ostrovskii
E-mail: osasha@ocean.ru
Mobile phone: +7-916-4905969